

# Optical, physical and chemical properties of aerosols transported to a coastal site in the Western Mediterranean : Focus on primary marine aerosols

Marine Claeys[1], Greg Roberts[1,2], Marc Mallet[1], Jovanna Arndt[3], Karine Sellegri[4], Jean Sciare[5,6], John Wenger[3], and Bastien Sauvage[7]

[1]CNRM, Centre National de Recherches Météorologiques UMR 3589, Météo-France/CNRS, Toulouse, France
[2]Scripps Institution of Oceanography, Center for Atmospheric Sciences and Physical Oceanography, La Jolla, United States
[3]Department of Chemistry and Environmental Research Institute, University College Cork, Cork, Ireland
[4]LaMP, Laboratoire de Météorologie Physique CNRS UMR6016, Observatoire de Physique du Globe de Clermont-Ferrand, Université Blaise Pascal, Aubière, France
[5]LSCE, Laboratoire des Sciences du Climat et de l'Environnement, Unité Mixte CEA-CNRS-UVSQ, CEA/Orme des Merisiers, 91191 Gif-sur-Yvette, France
[6]Energy Environement Water Research Center, The Cyprus Institute, 2121 Nicosia, Cyprus
[7]LA, Laboratoire d'Aérologie, Observatoire Midi-Pyrénées, CNRS/IRD/Université de Toulouse, 14, Avenue Édouard Belin, 31400 Toulouse, France

*Correspondence to:* M. Claeys (marine.claeys@cnrm-game-meteo.fr)

**Abstract.**

As part of the ChArMEx-ADRIMED campaign (summer 2013), ground-based in-situ observations were conducted at the Ersa site (northern tip of Corsica; 533 m asl) to characterize the physical, optical and chemical properties of aerosols. During the observation period, three different aerosol regimes have been identified, including a dust outbreak (Dust) originating from

5   Algeria/Tunisia, a primary marine aerosols (PMA) event from both the Gulf of Lion and emissions near the sample site, and a pollution period from Eastern Europe, which includes anthropogenic and biomass burning sources (BBP). The chemical, physical and optical properties of the observed aerosols as well as their local shortwave (SW) direct radiative forcing (DRF) in clear-sky conditions are compared for these three periods in order to assess the direct radiative impact of PMA above the Western Mediterranean Basin.

10   The PMA period is characterized by a mean sea salt mass concentration up to 6.5 $\mu$g m$^{-3}$, representing 40% of the total PM$_{10}$ mass concentration, and a relatively low ratio of chloride to sodium (average of 0.57) indicating a generally 'aged' sea salt aerosol at Ersa. In this work, an original dataset, obtained from on-line real-time instruments (ATOFMS, PILS-IC) have been used to characterize the ageing of PMA. The majority of PMA had surprisingly undergone chemical reactions and were mostly advected from long-range transport. During PMA period, the mixing between fresh and aged PMA originated from

15   both local and regional (Gulf of Lion) emissions.

The aerosol optical properties, obtained for the whole atmospheric column and at the surface, indicate a single scattering albedo (SSA) near unity (at 440 nm), indicating almost purely scattering particles, associated to a relatively low aerosol optical depth (AOD) close to 0.1 (at 500 nm), and an aerosol angstrom extinction exponent (AE) equal to 1.3 ± 0.4 (between 440 and





870 nm), suggesting a possible mixing of the PMA with fine particles (probably of continental origin). AERONET retrievals indicate a relatively low local SW DRF during the PMA period with mean values of -11±4 W m$^{-2}$ at the surface and -8±3 W m$^{-2}$ at the top of the atmosphere (TOA).

In comparison, our results indicate that the dust outbreak observed at our site during the campaign, although of moderate intensity (AOD of 0.3-0.4 at 440 nm and column-integrated SSA of 0.90-0.95), induced a local instantaneous SW DRF nearly three times the forcing calculated during the PMA period, with maximum values up to -40 W m$^{-2}$ at the surface. On average, the SW DRF was about -21±11 and -14±6 W m$^{-2}$, at the surface and at TOA, respectively, during this dust outbreak.

Finally, the BBP period was characterized by a significant increase of the aerosol PM1 mass concentration (from 3.7 $\mu$g m$^{-3}$ to 7.2 $\mu$g m$^{-3}$) due to the influence of biomass-burning and anthropogenic aerosols transported from Eastern Europe. The influence of polluted/smoke particles led to a significant decrease in SSA (0.90 at 440 nm), showing the important absorbing characteristics of such particles. For this period, the SW DRF at the surface and TOA also exhibit higher mean values compared to the PMA period (with values of -23±6 W m$^{-2}$ and -15±4 W m$^{-2}$, respectively) and similar range of values as the Dust period.

# 1 Introduction

The Mediterranean Basin is a crossroad for air masses bringing different types of aerosols, both from natural and anthropogenic origins (Lelieveld et al., 2012). Among these aerosols, primary marine aerosols (PMA) (mainly composed of sea salt and to a lesser extent of organic matter) are important because they are always present over the Mediterranean basin and compose the main part of background aerosols over the Mediterranean (Pace et al., 2006). They are able to chemically react with other aerosol species, act as cloud condensation nuclei (CCN) and also interact with solar and thermal radiations due to their large size range (Kaufman et al., 2001; Kaufman and Koren, 2006; Khain, 2009; Li et al., 2011; Seiki and Nakajima, 2014). At the global scale, Bellouin et al. (2008) estimated that the contribution of marine aerosols was equivalent to half of the total Direct Radiative Forcing (DRF), while Zhao et al. (2011) found a contribution of one third of the total DRF. However, their contribution is highly variable in time and space, due to spatial variations in wind speed and pre-existing particle loadings. At the regional scale and over the Mediterranean basin, Salameh et al. (2007) indicate that the contribution of sea-salt particles to the total aerosol loading and optical depth ranges from 1 to 10 %. They report Aerosol Optical Depth (AOD) around 0.15-0.20 (at 865 nm) within the sea salt aerosol plume during strong wind (e.g., Mistral and Tramontane) events. In addition, Mulcahy et al. (2008) reported a high correlation between AOD (at 500 nm) and wind-speed, with AOD values of 0.3-0.4 at moderately-high wind speed - most likely related to the increase in PMA. Consequently, the persistent and punctually elevated AOD due to PMA aerosols can have an important impact on the radiative budget of the Mediterranean basin. This high variability in terms of PMA loading and physical, chemical and optical properties leads to important uncertainties in the quantification of regional radiative impact, both for direct and indirect effects (Forster et al., 2007; Stevens and Feingold, 2009). Finally, it should be noted that most past studies have documented aerosol properties in the Eastern part of the Mediterranean Basin (Crete (Mihalopoulos et al., 1997; Bardouki et al., 2003; Sciare et al., 2003; Koulouri et al., 2008); Greece (Chabas and Lefèvre, 2000)).



In that context, the aim of this study is to characterize the chemical, physical and optical properties of PMA compared to the other major aerosol sources affecting the Western Mediterranean basin.

This work has taken place in the frame of the ChArMEx-ADRIMED (Chemistry Aerosols Mediterranean Experiment - Aerosol Direct Radiative Impact on the regional climate in the MEDiterranean region) project (https://charmex.lsce.ipsl.fr)
that took place in the Western Mediterranean basin during the summer 2013 (Mallet et al., 2016), and we used the real-time measurements performed at the remote ground-based Ersa atmospheric station situated at Cape Corsica (42.9694° N, 9.3803° W, altitude of 533m asl.).

The first part of this study (Sect. 2) describes the instrumentation deployed at the Ersa station and the FLEXPART model configuration used to identify air masses origins at the station. Periods of the field campaign affected by the major aerosol
sources are then discussed (Sect. 3.1 and 3.3) using chemical and physical measurements, as well as back trajectory analysis and direct radiative forcing. Finally, meteorological observations recorded near the sample site (Ersa) during a period characterized by a particularly strong source of PMA is used to adress the dependence of PMA mass concentration and ageing to local and regional wind speed (Sect. 3.2.2).

## 2 Method

### 2.1 Atmospheric station and instrumental set-up

The research atmospheric station of Ersa is located at the top Northern part of Corsica Island (Cape Corsica; 42.9694° N, 9.3803° W). Its altitude is 533 m above sea level (asl), and it is surrounded by the Mediterranean sea on its northern, eastern and western sides and by mountains ($\simeq$ 1000 m asl) on its southern side. The station is located in a remote area, with minimal influence of local anthropogenic emissions. A more detailed description of the station is given by Mallet et al. (2016).

This station is equipped to provide in-situ measurement of the aerosol physical properties, including number concentration and number size distribution, using a Scanning Mobility Particle Sizer (SMPS 3081, TSI INC.), an Optical Particle Sizer (OPS 3330, TSI INC.) and an Aerodynamic Particle Sizer (APS 3321, TSI INC.) to characterize both submicron and supermicron aerosol particles. Aerosol size distributions was achieved using two sampling inlets. The first one was a $PM_{10}$ head inlet, in which the air flow was dried using a Nafion dryer (TSI INC.) to a relative humidity below 40% and then divided into several
paths to the OPS, SMPS and a Condensation Particle Counter (CPC 3010, TSI INC.). The second head inlet was a $PM_{20}$ with a flow rate of 1L.min$^{-1}$ that sampled for the APS. The flow rate reaching the other instruments was 1L.min$^{-1}$ for the OPS and 0.5 L.min$^{-1}$ for the SMPS. The CPC measured the total number aerosol concentration for electric mobility diameters larger than 10 nm. The SMPS counted the number of particles per size bins from 10 to 500 nm, while the APS measured at ambient RH the number of particles per size bins from 0.5 to 20 $\mu$m and the OPS from 0.3 to 10 $\mu$m. Particulate matter below 1 and 10
$\mu$m ($PM_1$ and $PM_{10}$ respectively) were measured at the station on hourly basis using a TEOM-FDMS (Thermo Environment, model 1405-F) and a TEOM (Thermo Environment, model1400), respectively.

Aerosol optical instruments were also deployed (nephelometer TSI INC. at 3 wavelengths, MAAP) to determine light scattering and absorption properties of aerosols. The nephelometer measured the scattered and backscattered coefficients at three



wavelengths, 450 (blue), 550 (green) and 700 nm (red) with a $PM_{10}$ head inlet (flow rate of 40L.min$^{-1}$), while the MAAP instrument (Multi Angle Absorption Photometer, Thermo Scientific) measured the concentration of black carbon from the absorption of particles at the 670 nm wavelength.

The PILS-IC measurements were performed using a Particle-into-Liquid-Sampler (PILS, Orsini et al. (2003)) running at 16.8 ($\pm$ 0.5) LPM and coupled with two Ion Chromatographs (IC) for the determination of the major cations and anions. More details on this instrumentation and its comparability with other real-time aerosol analysers can be found in Zorn et al. (2008); Sciare et al. (2011); Healy et al. (2013); Crippa et al. (2013); Bressi et al. (2013). Basic and acidic annular denuders (3-channel, URG Corp., USA) were mounted upstream of the PILS instrument and downstream of a PM10 inlet having a 50% cut-off diameter of 10 $\mu$m at 16.67 LPM. Ambient concentrations of ions were corrected from blanks performed every day for 1h and achieved by placing a total filter upstream of the sampling system. Liquid flowrates of the PILS were delivered by peristaltic pumps and set to 1.5ml/min for producing steam inside the PILS and 0.37 ($\pm$ 0.02) ml/min for rinsing the impactor. Cation measurements were performed using an IC (Dionex, model ICS1100) equipped with a 2-mm diameter Auto-Suppression, Cation Self-Regenerating Suppressor (CSRS), a 2-mm diameter CS-12 pre-column and column, and a 100 $\mu$l injection loop. Analyses were performed in isocratic mode at 20mM of Methanesulfonic Acid (MSA) at a flowrate of 0.25ml/min, for the quantitative determination of the 5 major cations ($Na^+$, $NH_4^+$, $K^+$, $Mg^{2+}$, $Ca^{2+}$) every 12 min. Based on these IC settings, the detection limit ($2\sigma$) for cations was typically 0.1 ppb, which corresponds to an atmospheric concentration of $\sim$ 1 ng/m$^3$. Calibration was performed every 2 weeks for concentrations ranging from 10 to 800 ppb and showed a drift below 5% for each cation between the beginning and the end of the campaign. Anion measurements were performed using an IC (Dionex, model ICS2000) equipped with a 2-mm diameter Auto-Suppression, Anion Self-Regenerating Suppressor (ASRS), a 2-mm diameter AS-11 HC pre-column and column, and a 500 $\mu$l injection loop. Analyses were performed in isocratic mode at 10mM of KOH at a flowrate of 0.25ml/min, for the quantitative determination of the 5 anions (methanesulfonate, $Cl^-$, $NO_3^-$, $SO_4^{2-}$, oxalate) every 24 min. Based on these IC settings, the detection limit ($2\sigma$) for anions was typically 0.1 ppb, which corresponds to an atmospheric concentration of $\sim$ 1 ng/m3. Calibration was performed every 2 weeks for concentrations ranging from 10 to 800 ppb and showed a drift below 5% for each anion between the beginning and the end of the campaign. To our best knowledge, this is the first time that PILS-IC measurements are reported in $PM_{10}$, providing here a unique opportunity to document water-soluble supermicron ions and sea salt in particular. Quality control of the PILC-IC data was successfully performed by comparison with $PM_{10}$ filter (Teflon)-based ion measurements performed in parallel on a 12-h time basis (Leckel, SEQ47/50 model running at 2.3 m$^3$/h), with typically less than 20% discrepancies for the major anions/cations.

PMA concentration was calculated using these data and the following formula (Brewer, 1975) :

[PMA]=$[Cl^-] + [Na^+] + [ss\text{-}Mg^{2+}] + [ss\text{-}SO_4^{2-}] + [ss\text{-}Ca^{2+}] + [ss\text{-}K^+]$ where [ss-X] / $[Na^+]$ = 0.13, 0.251, 0.039 and 0.036 corresponding to $Mg^{2+}$, $SO_4^{2-}$, $Ca^{2+}$ and $K^+$ respectively. The ACSM measured the chemical composition of non-refractory $PM_1$ (Organic matter (OM), Nitrate ($NO_3$), Sulfate ($SO_4$), Ammonium ($NH_4$), Chloride (Cl)) with a time resolution of 30 min. The chemical composition of non-refractory submicron aerosol has been continuously monitored using a Quadrupole Aerosol Chemical Speciation Monitor (Aerodyne Research Inc.), which has been described in detail by Ng et al. (2011). Briefly, $PM_{2.5}$ aerosols are sampled at 3 L/min (from a $PM_{2.5}$ cyclone inlet) and then sub-sampled at 85 mL/min (volumetric flow) through





an aerodynamic lens, focusing submicron particles (40-1000 nm aerodynamic diameter, A.D.) onto a 600 C-heated conical tungsten vaporizer where non-refractory material is flash-vaporized and quasi instantaneously ionized by electron impact at 70 eV. Briefly, the instrument calibration has been performed following the recommendation of Jayne et al. (2000) and Ng et al. (2011), where generated mono-disperse 300 nm A.D. ammonium nitrate particles are injected into both ACSM and a

condensation particle counter (CPC) at different concentrations. It has been successfully intercompared against 15 other aerosol mass spectrometers (Crenn et al., 2015; Fröhlich et al., 2015). Quality control of ACSM data was successfully performed by comparison of $PM_1$ (sum of chemical species measured by ACSM and MAAP) with $PM_1$ obtained with SMPS (with density of 1.5).

The ATOFMS (aerosol time-of-flight mass spectrometer), deployed by University College Cork, measured the vacuum aero-

dynamic diameter of the individual particles and their chemical composition. A detailed description of the ATOFMS (TSI INC. model 3800) can be found elsewhere (Gard et al., 1997). Briefly, it consists of an aerodynamic focussing lens (TSI AFL100) (Su et al., 2004) that transmits particles in the diameter range 100-3000 nm, a particle sizing region, and a bipolar reflection time-of-flight mass spectrometer. Single particles are desorbed/ionized using a pulsed Nd:YAG laser ($\lambda$ = 266 nm, $\simeq$1 mJ.pulse$^{-1}$). Positive and negative ion mass spectra of individual aerosol particles are obtained which enable identification

of the chemical constituents. The AFL reduces the transmission efficiency of supermicron particles, while variability in the desorption/ionisation laser influence results in qualitative mass spectral signals.

The aerosol optical properties were retrieved from the AERONET/PHOTONS network. We used here the level 1.5 data obtained from the sun-photometer located near Ersa station http://aeronet.gsfc.nasa.gov/cgi-bin/type_one_station_opera_v2_

new; the Aerosol Optical Thickness (AOD) derived at 8 wavelengths (from 340 to 1640 nm), the angstrom exponent (AE) was calculated using the AOD at 440 and 870 nm, and the volume size distribution was retrieved from the algorithm proposed by Dubovik et al. (2002b). The Single Scattering Albedo (SSA) provides crucial informations related to the ratio of scattering to extinction (scattering plus absorbing) of radiations by aerosols. The sun-photometer data are available several times per day, depending on the solar angle and aerosol loading (Dubovik et al., 2002b). We have also used the aerosol clear-sky instantaneous

direct radiative forcing in the shortwave derived from sun-photometer measurements, following the methodology proposed by García et al. (2012).

In addition to the whole atmospheric column observations of optical properties, the TSI nephelometer deployed at Ersa was used to determine the scattered light at three different wavelengths, blue (450nm), green (550nm) and red (700nm). This

instrument provides the scattering coefficient (not directly linked to the concentration of particles), associated to an indication of the size of aerosols through the spectral dependence of the scattering coefficient between two wavelengths.

Temperature, relative humidity, wind speed and direction measured in real-time during the whole campaign were taken from the Ersa atmospheric station. Because the wind measurements may have been influenced by the orography around the station, we used the wind data provided by the closest Météo-France station (Semaphore station), which was situated about 5 km away



from Ersa and close to the sea. For the sequence of the study, when mentionned local wind measurements, it refers to wind observed at the Semaphore station.

## 2.2 ATOFMS data analysis

The distinction between aged and fresh sea salt was performed according to the detection of chloride and nitrate in the particles.
For fresh sea salt, we obtained signals for various chloride ions ($^{81,83}Na_2Cl^+$, $^{35,37}Cl^-$ and $^{93,95}NaCl_2^-$) and also some nitrate ($^{46}NO_2^-$, $^{62}NO_3^-$). The signals for chloride were lower and those for nitrate stronger for aged sea salt, due to the replacement of chloride and sodium nitrate formation (Noble and Prather, 1997; Gard et al., 1998). In this case, the relatively small signals for chloride ions is a good indicator of aged sea salt aerosol. The size distribution of aged and fresh sea salts was investigated using this differentiation. Regarding the size distribution, the main limitation is the upper cut-off size of 3 $\mu$m which limits the detection mostly to fine sea salt particles.

Average mass spectra for aged and fresh are shown in Fig. 1. Both sea salt classes are typical of those observed in other coastal/marine environments (Gard et al., 1998; Dall'Osto et al., 2004; Healy et al., 2010). The positive modes for both fresh and aged particles are similar and are characterised by sodium ions ($^{23}Na^+$, $^{46}Na_2^+$, $^{62}Na_2O^+$, $^{63}Na_2OH^+$ and $^{81,83}Na_2Cl^+$) and $^{39}K^+$. The negative mass spectra for fresh sea salt particles shows peaks for $^{16}O^-$, $^{35,37}Cl^-$, nitrate ($^{46}NO_2^-$, $^{62}NO_3^-$) and $^{93,95}NaCl_2^-$, while the signals for nitrate dominate the aged sea salt negative mode and sodium chloride adducts are virtually absent. The occurrence and relative intensity of the chloride and NaCl adducts in ATOFMS mass spectra are key markers for distinguishing fresh sea salt from aged sea salt. The absence of NaCl ions and strong nitrate signals indicates extensive replacement of Cl by NO3, while the presence of nitrate in the negative mass spectra of the fresh sea salt particles suggests that these are not truly fresh but have also undergone some Cl replacement.

## 2.3 FLEXPART model

We used the FLEXible PARTicle (FLEXPART) Lagrangian dispersion model (Stohl et al., 1998), version 9.02. FLEXPART in a backward mode during the campaign to identify the sources and the transport time of air masses observed at the Ersa station. The model is driven by wind fields provided by the European Centre for Medium-range Weather Forecast (ECMWF) using both analyses and forecasts with a temporal resolution of 3 hours (00, 06, 12, 18h UTC for analyses and 03, 09, 15, 21h UTC for forecast). The horizontal resolution is 0.141° × 0.141° and 91 vertical levels are used (137 after 25 June 2013). Turbulence is parameterised solving Langevin equations (Stohl and Thomson, 1999) and the convection parameterisation scheme is adopted from Emanuel and Živkovic-Rothman (1999) for all types of convection. The model calculates trajectories of user-defined ensembles of particles released from a three-dimensional box in backward mode during 6 days. In this study $10^4$ particles were released at the beginning of each run in a 100 km × 160 km × 200 m (lat × lon × alt) box centred above Cap Corsica (Northern tip of Corsica Island). Backtrajectories were performed for 3 different altitudes : 500 m (bottom and top of the box at 400 m and 600 m respectively) corresponding to the altitude of the measurement site at Ersa, 2,000 m corresponding to an altitude above the boundary layer and 4,000 m corresponding to an altitude where the ATR 42 research aircraft mainly observed dust plumes (Mallet et al., 2016). Besides the particles' positions, FLEXPART also includes cluster analysis for





particle ensembles (Stohl et al., 2002) and the average residence time of particles in the output grid cells. Cluster analysis uses the plume dispersion information (residence time) to calculate, at each time step (3 hours), 10 clusters, (using the k-means clustering) synthetizing the particles dispersion information. All the particles are contained and allocated in these 10 clusters according to their position (latitude, longitude and altitude (Z)). The horizontal resolution for the Flexpart output grid was 1°

× 1° and the vertical resolution was 500 m from the ground up to 9500 m.

### 2.4 Aerosol mass closure

In order to assess the consistency of the chemical data set, we have compared the TEOM $PM_{10}$ and $PM_1$ data with the on-line chemical concentration measurements performed in parallel. $PM_{10}$ mass concentrations were compared to the sum of chemical components obtained from the PILS-IC $PM_{10}$ data, the BC concentration from the MAAP instrument ($PM_{2.5}$), and the organic

matter (OM) concentration derived from the ACSM instrument ($PM_1$). We note that the reconstructed mass underestimates the TEOM $PM_{10}$ concentration by a factor ranging from 0.5 to 1 with a poor correlation coefficient ($r^2$=0.31). This lack of aerosol mass could be due to the mass of (insoluble) dust not determined chemically or possibly a supermicron mode of organic that was not determined here.

For this study, the Sulfate ($SO_4^{2-}$) and $NH_4^+$ data were taken from PILS-IC instrument as the correlation between ACSM

and PILS-IC measurements show a very good agreement ($SO_4^{2-}$(PILS) = 0.99 × $SO_4^{2-}$(ACSM)), $r^2$ = 0.95 and $NH_4^+$(PILS) = 1.27 × $NH_4^+$(ACSM), $r^2$ = 0.87).

## 3 Results

### 3.1 Overview of aerosols sources

In this section, the chemical properties of the aerosols measured in Ersa (Fig. 2) is first studied, revealing a significant variability

in the contribution of the different aerosol species and outlined three mains periods, Dust, PMA and BBP, under the influence of different types of air masses and particles. The aerosol physical properties are then discussed in Sect. 3.2.3 and 3.3.1.

The mean $PM_{10}$ concentration measured by the TEOM $PM_{10}$ during the ADRIMED campaign was $11.5 \pm 5.4 \ \mu g \ m^{-3}$. For the majority of the sampling period the mass concentration ranged from 10 to 20 $\mu g \ m^{-3}$, except for short periods when the concentration falls to 5 $\mu g \ m^{-3}$. These decreases are usually due to wet scavenging or to the diurnal variation of the boundary

layer, as the Ersa station was within the boundary layer during daytime and sometimes slightly above the boundary layer during night-time (aerosol concentrations at night were often lower when the Ersa site was in the free troposphere). In parallel, the mean $PM_1$ concentration measured by the TEOM $PM_1$ during ADRIMED was $6.4 \pm 3.2 \ \mu g \ m^{-3}$. The concentration was lower during June and rises during the beginning of July to exceed 10 $\mu g \ m^{-3}$.

The major chemical constituents of $PM_{10}$ measured at Ersa (Fig. 2) show a significant temporal variability during the campaign.

Three main periods under the influence of different types of air masses and aerosols have been selected here and discussed in more details below.



The first period (16 to 20 June) corresponds to a dust outbreak and is characterised by the concentration of non-sea-salt Calcium (nss-$Ca^{2+}$) concentration, a proxy of desert dust (Sciare et al., 2003), which increases from 0.5 to 2 $\mu$g m$^{-3}$. This dust event lasted a few days, from 16 to 20 June, with nss-$Ca^{2+}$ concentrations peaking on 18 June at 2 $\mu$g m$^{-3}$ at the Ersa site. In addition, the concentrations of calcium measured by the PILS-IC are relatively low for a dust event, because the maximum

concentration of dust particles was located at an altitude ranging between 3 to 6 km (Denjean et al., 2016). The second reason concerns the method used by the PILS-IC instrument that analyzes only the soluble fraction of aerosols, while a significant part of dust Ca is insoluble. In that sense, the concentration of nss-$Ca^{2+}$ determined by PILS-IC remains a qualitative indicator of the presence of dust particles.

PILS-IC measurements also indicate an increase in the concentrations of Oxalate, potassium, $SO_4^{2-}$ and $NH_4^+$ from 5 to 9

July, which corresponds to BBP influences. A brief increase of $NO_3^-$ concentrations (PILS-IC) was recorded on 5 and 6 July (Fig. 2).

This event was also detected by the ACSM. Indeed, there is a difference of a factor of two in terms of total mass concentrations between the first part of the campaign (6 June to 4 July) and during the BBP period. During the first period (6 June to 4 July), the mass concentration of each aerosol species is low, characterized by a mean total concentration of 3.7 $\pm$ 1.6 $\mu$g m$^{-3}$.

During the second identified period (4 to 13 July), the total PM$_1$ aerosol mass concentration increases suddenly to reach a mean of 7.2 $\pm$ 1.7 $\mu$g m$^{-3}$. Similarly, the total PM1 mass concentration measured by the TEOM increases from 5.9 $\pm$ 3.0 to 8.4 $\pm$ 3.3 $\mu$g m$^{-3}$. This increase is mainly due to a large addition of the concentration of submicronic organics compounds (from 2.1 $\pm$ 0.9 $\mu$g m$^{-3}$ to 4.1 $\pm$ 1.2 $\mu$g m$^{-3}$) and an increase of $SO_4^{2-}$ (and $NH_4^+$) from 0.9 $\pm$ 0.6 (0.5 $\pm$ 0.3) to 1.8 $\pm$ 0.6 (0.9 $\pm$ 0.3) $\mu$g m$^{-3}$. Organics, sulfate and ammonium concentrations remain high until 10 July after which they decrease, but to values

still higher than during the month of June. In parallel, the black carbon (BC) concentration is found to be low during the whole period of the campaign, although we observe an increase during July (mean of 0.41 $\pm$ 0.11 $\mu$g m$^{-3}$) compared to June (mean of 0.28 $\pm$ 0.11 $\mu$g m$^{-3}$). For a few days and during the PMA period, the concentration of BC is found to be very low (0.20 $\pm$ 0.09 $\mu$g m$^{-3}$) and recover its previous concentration until 3 July. The highest BC concentration (0.75 $\mu$g m$^{-3}$) was reached on 5 July. The ACSM observations clearly indicate that concentrations of all the chemical components during the BBP episode

are twice the concentration they had during the first period of the ADRIMED campaign. Similar episods of european origin and biomass burning events were studied by Ripoll et al. (2015) in a Mediterranean site, with an increase of the concentration of PM$_{10}$ nitrate (6 times higher than the annual average), sulfate (about 3 times higher than the annual average), ammonium (more than 4 times higher than the annual average), OM (about 2 times higher than the annual average) and potassium (2 times higher than the annual average).

Our observations reveal that the mean concentration of inorganic PMA (averaged for the months of June and July 2013) was found to be low with a value of 0.76 $\pm$ 1.04 $\mu$g m$^{-3}$ (Fig. 2). However, during PMA period, the concentration of PMA increases up to 6.5 $\mu$g m$^{-3}$, with a mean concentration of 3.2 $\pm$ 1.8 $\mu$g m$^{-3}$. For this specific period, the mass of PMA represents, on average, 22 % of the total mass measured by the TEOM PM$_{10}$ instrument, while the average contribution was about 7 % for the whole period of observations. At the Ersa station, the highest concentration of PMA was reached on 24 June

(Fig. 3), when PMA concentration represents 40 % of hourly PM$_{10}$ mass concentration for 25 % of the data. Even though the





mean values of PMA mass concentration measured at Ersa were low compared to values referenced at other Mediterranean sites, this contribution still remains significant. Indeed, Pey et al. (2009) reported a ratio of 10 % of sea spray (sum of Na and $Cl^-$ mass concentrations from Quartz fibre filter) to $PM_{10}$ (annual mean = 2.9 $\mu g\ m^{-3}$) in Mallorca Island (117 m.asl), while Bardouki et al. (2003) found a contribution of 40% of PMA in the coarse mode of inorganic ions during summer at Finokalia

(150 m. asl, Crete Island). Querol et al. (2009) analysed the chemical composition of $PM_{10}$ aerosols in the Mediterranean Basin and found a mean annual contribution of sea spray to $PM_{10}$ that did not exceed 24%. So the contribution of inorganic sea salt to the $PM_{10}$ mass concentration in the Mediterranean Basin is on average lower than 20% but can reach 40% during particular events such as the one observed at Ersa in June 2013. Moreover, sea spray likely comprises a substantial fraction of organic PMA and hence may represent a larger fraction of $PM_{10}$ than the one estimated solely from the inorganic fraction (Gantt and

Meskhidze, 2013).

### 3.1.1 Origins and time of residence of the different air-masses observed at Ersa

The origin of air masses impacting Ersa for the three different periods depicted in Fig. 2 has been investigated using FLEXPART model in order to characterise the transport time and the emitting sources of these aerosols.

Figure 5a represents the time series computed from FLEXPART clusters backtrajectories. The upper one represents the

transport of the air masses passing over different regions before reaching the Ersa station. These different zones take into account the regions influenced by anthropogenic pollution, biomass burning and marine influences and are represented in Fig. 4. They represent the most probable influence on air masses arriving in Cap Corsica due to their close locations and specific emissions.

In general, the Ersa station was influenced by air masses coming from West and South during the first part of the field

campaign (from 6 to 26 June), and was more influenced by air masses coming from East and North during the last part of the campaign (26 June to 13 July). During the campaign, Ersa was always affected by air masses that passed over French or Italian coastal areas. The influence of the Mediterranean coasts of Spain is also very present during the campaign, especially in the first part of June.

In terms of transport time, FLEXPART simulations indicate that air masses spent a few hours to several days over the sea

after leaving the French or Italian coasts, and from 2 to 6 days from continental European sources. Coastal regions are likely the source of pollution-anthropogenic impacted air masses, because they are highly industrialized and populated and are the last major source of anthropogenic aerosols before transport over the Mediterranean Sea. At the local scale, Ersa was mostly under the influence of a westerly wind ($\simeq 270°$) (Fig. 2) from the beginning of the campaign to the beginning of July, expect for a few days during the dust outbreak (from 16 to 20 June), where it was under a South-Eastern influence ($\simeq 150°$). Finally,

from 4 to 9 July, Ersa was experiencing mostly an Easterly wind ($\simeq 100°$).

The influence of Southerly air masses is marked by the passage of air masses above North Africa, at the beginning of the campaign (19 June, Fig. 6). The transport time of air masses from Northern Africa to Ersa ranges between 2 to 6 days. Such air masses contain important concentrations of mineral dust particles, which are usually transported at higher altitudes over the Mediterranean basin, in the free troposphere and up to 9 km in altitude (Denjean et al., 2016; Hamonou et al., 1999; Dulac



and Chazette, 2003; Di Iorio et al., 2009; Mona et al., 2006; Gómez-Amo et al., 2011). Thus, we also performed simulations starting at 4000 m asl., that show that the air masses arriving at Cap Corsica on 19 and 20 June were within the boundary layer (< 1000 m) over Tunisia and Algeria from 2 to 3 days before. At the Ersa site, during the dust outbreak, around 19 June, the wind speed reached 15 m s$^{-1}$.

Besides the coastal anthropogenic influence observed during the first week of July, Fig. 6 shows that the air masses came from Eastern Europe from 7-12 July and in particular from Ukraine 3-4 days before reaching Ersa. The date at which the air masses passed over these regions correspond to important emissions of biomass burning observed near the Black Sea as shown by the MODIS satellite retrievals (http://rapidfire.sci.gsfc.nasa.gov/cgi-bin/imagery/firemaps.cgi).

FLEXPART back-trajectory simulations also show that during the PMA period, air masses were coming from the North-
West of Cap Corsica, including Gulf of Lion. This is consistent with a higher PMA concentration in Ersa, as a longer fetch leeds to higher mass concentration. Our simulations reveal that these air masses were also influenced by anthropogenic sources from France and Italy. The study of the transport of air masses passing over maritime zones (Fig. 5b), and especially Gulf of Lion and North Atlantic zones gives us information about the transport time from the source regions to Ersa and their altitude. Our simulations indicate that the mean transport time from the Gulf of Lion is less than a day for the whole period except for
the last day, 26 June. Whereas from the North Atlantic zone (Bay of Biscay), the transport time is at least more than 2.5 days and increases up to 4.5 days for 26 June (Table 1). Almost no precipitation occurred during this period between the Bay of Biscay and Corsica, so these air masses were likely not impacted by wet scavenging.

The mean altitude for the air masses coming from Gulf of Lion is close to 1000 m (972 m ± 753) for the 5 days, with minima mainly below 500 m, while the mean altitude from the North Atlantic is 1374 m (± 828) and the minimum being below 800
m only for the first 3 days. Tsyro et al. (2011) reported that the concentration of sea salt aerosols associated to emissions was highest until altitudes of 600-700 m, which correspond typically to the Marine boundary layer (MBL) height. Thus an influence from the North Atlantic Ocean would occur more likely during the first 3 days of the period, when the air masses lay within the MBL. Concerning the Gulf of Lion, the altitudes of the air masses are low enough to bring sea salt aerosols in Ersa. While vertical transport is not well captured in the model, FLEXPART model indicates that most of the PMA aerosol mass is
transported in the MBL.

To summarize, our FLEXPART simulations clearly indicate that the Ersa site was impacted by a disperse set of air masses coming from different regions transporting different types of aerosols, and these FLEXPART results are consistent with chemical measurements obtained at Ersa station, as well as the three periods discussed here.

The following sections will focus on the physical, chemical and optical properties of aerosols sampled during the PMA
period. The Dust and BBP events will be used as a comparison for different states of the atmosphere impacting the Ersa site.



## 3.2 Primary marine aerosols

### 3.2.1 PMA ageing

As reported by Clegg and Brimblecombe (1985) and Quinn and Bates (2005), the ratio of the concentration of $Cl^-$ over $Na^+$ is an indicator of the chloride depletion that happens when PMA react with acidic gases like $HNO_3$ and $H_2SO_4$ according to the chemical reactions R1, R2 and R3:

$$H_2SO_4 + 2NaCl \rightarrow Na_2SO_4 + 2HCl \qquad (R1)$$

$$NaCl + H_2SO_4 \rightarrow NaHSO_4 + HCl \qquad (R2)$$

$$NaCl + HNO_3 \rightarrow HCl + NaNO_3 \qquad (R3)$$

These reactions result in a loss of particulate chloride in PMA during transport. The typical mass ratio of $Cl^-/Na^+$ of the sea water is 1.8 (Lewis and Schwartz, 2004); however, the study of $PM_1$ PMA in the Mediterranean basin by Schwier et al. (2016) shows a $Cl^-/Na^+$ ratio of 1.2. Numerous values are referenced over the Mediterranean basin; 0.6 for long-term measurements (July 2012 - April 2013) in Ersa station (Nicolas, 2013), 0.49 by Mihalopoulos et al. (1997), 1.00 by Koulouri et al. (2008) and 1.2 during summer by Bardouki et al. (2003) in Finokalia (Eastern Mediterranean, Crete Island) and 1.2 during summer in the Eastern Mediterranean coast of Turquey by Koçak et al. (2004). These values are found to be low compared to the seawater ratio, especially for Mihalopoulos et al. (1997), probably related to the high reactivity of chloride with acidic gases that are present in relatively high concentrations in the Mediterranean atmosphere (Sellegri et al., 2001; Bardouki et al., 2003; Pey et al., 2009). A good correlation was found between $Na^+$ mass concentration and the sum of $Cl^- + NO_3^-$ mass concentrations (PM10 measurements) ($r^2 = 0.87$) indicating that $NO_3^-$ is the main component interacting with sea salt.

During PMA period, the $Cl^-/Na^+$ mass ratio varies between 0.13 and 1.3 (Fig. 9), with a mean of $0.59 \pm 0.23$. This result is consistent with the long term measurement performed between July 2012 and April 2013 at Ersa (Nicolas, 2013). This indicates that PMA measured in Ersa (and throughout the Mediterranean Basin) were predominantly aged.

To distinguish 'mostly aged' and 'mostly fresh' PMA, we used a spectral analysis of the ATOFMS measurements. The terms fresh and aged PMA that will be used from now in this text correspond to the classification made with the ATOFMS. During the ChArMEx-ADRIMED campaign, an alternation between these two states of PMA was detected.

The size distribution of these two ATOFMS sea salt types were fitted according to a sum of lognormal modes. The fresh PMA were characterised by one mode with a vacuum aerodynamic diameter of 1.29 $\mu$m and a standard deviation $\sigma$ of 1.34, while the aged PMA were characterised by 3 different modes, as detailed in Table 2.



Our results show that during the campaign aged PMA are dominant, but during PMA period (22-26 June), when the wind near Cap Corsica is higher (Sect. 3.2.2), there is an alternation of short events of fresh or aged PMA, with a dominance of fresh PMA. The comparison of the ATOFMS and PILS data show a relatively good agreement between the two instruments regarding the dominance of fresh and aged PMA (Fig. 9).

To compare the two instruments, we looked at the count ratio Aged/Fresh PMA, and attributed a state to the $Cl^-/Na^+$ ratio measured by the PILS-IC. For a large number of measurements only aged aerosols were detected and attributed as 'Only aged' state. When the count ratio of Aged aerosols over Fresh aerosols was higher than one, the measurements were characterised as 'Mostly Aged', and less than one the PMA were considered 'Mostly Fresh'. One can observe in Fig. 9 that the $Cl^-/Na^+$ ratio is higher when the ATOFMS distinguished fresh PMA, and lower when the ATOFMS distinguished aged PMA. We then

determined the mean $Cl^-/Na^+$ ratio for mostly aged ($0.38 \pm 0.15$) and mostly fresh PMA ($0.62 \pm 0.17$). In our observations, the mostly fresh PMA ratio remains low compared to the initial ratio of 1.8 (Lewis and Schwartz, 2004) or even 1.2 for $PM_1$ PMA (Schwier et al. (2016)), revealing that even though PMA are characterized as 'fresh', they have undergone chemical reactions before reaching Ersa station.

### 3.2.2 PMA sources

Complementary to the FLEXPART results, we used wind measurements at the Semaphore station, at the Gulf of Lion buoy and at the Bay of Biscay buoy to investigate the possible relationship between the increase in PMA concentration observed in Ersa and the wind speed at these stations, and better assess the origin of sea salt aerosols at Ersa.

During the ChArMEx/ADRIMED campaign, the majority of air masses analysed containing PMA were coming locally from the West and the concentration of marine particles increased with wind speed (Fig. 7a). The wind direction is constant around

270° for 6 days (21-26 June), and fluctuates afterwards between East and West origins. The maximum wind speed (20 m s$^{-1}$) encountered during the campaign was observed on 24 June, coinciding with the highest sea salt mass concentration measured.

To investigate the relationship between wind speed and concentration of PMA measured in Ersa, we averaged its concentration by wind bins of 1 m s$^{-1}$ for different cases. We first looked at the relationship between the concentration of PMA in Ersa and the wind speed measured at the Semaphore for the whole period of the campaign (Fig. 7a). The result indicates a

relationship between the wind speed and PMA concentration and the best fit ($r^2 = 0.92$) is presented in the form of *ln [PMA] = a × WS + ln(M$_0$)*, where WS corresponds to the wind speed, $M_0$ ($\mu g$ m$^{-3}$) to the concentration that corresponds to a wind speed WS=0. The error bars correspond to $2\sigma$ rms (root mean square). Above 13 m s$^{-1}$, the concentration starts to rise rapidly. The relationship described here is compared with fit parameters found by Bressan and Lepple (1985); Taylor and Wu (1992); Marks (1990) (Lewis and Schwartz, 2004) chosen because the time resolution of the measurements were similar to those in Ersa and

the wind speed encountered during their measurements were in the same range as in Ersa during the campaign. Despite the high correlation between PMA concentration and wind speed shown here, our results yield mass concentrations at least an order of magnitude less than other studies shown in Fig. 7. This difference is probably related to the sampling altitudes, which for our study was 533 m.asl, and $\sim$ 10 m asl above sea surface for Bressan and Lepple (1985); Taylor and Wu (1992) and Marks (1990). This is contrary to Fomba et al. (2014) who did not find a significant correlation between PMA concentrations



and wind speed in Cap Verde, even though they found an increase of PMA concentration on days of higher wind speeds. Sellegri et al. (2001) had difficulties to establish a relation between local emission of PMA and wind speed measurements using instrumentation with a long integration time during the FETCH campaign, in accordance with previous results of Quinn et al. (2000). Shinozuka et al. (2004) found that the wind speed was a good indicator for a measuring period but not for a specific case.

To investigate the origin of PMA in fonction of their ageing, we distinguished the air masses that contain fresh or aged PMA, using the method defined in Sect. 3.2.1, for the whole campaign. We observe that the concentration of aged PMA (Fig. 7b) is constant and does not depend on the local wind speed, which suggest that the Ersa site is always impacted by long range transport containing aged PMA, even if the concentration is low ($0.6 \pm 0.2$ $\mu$g m$^{-3}$). In the contrary, we observe that fresh PMA concentration measured at Ersa (Fig. 7b) is highly dependant of the wind speed, following a fit of the form *(*ln [PMA] = a $\times$ WS + ln(M$_0$) with a correct correlation (r$^2$= 0.59). This result indicates that the highest concentration of PMA measured in Ersa during the campaign corresponds to fresher aerosols, and is dependent on the local meteorological conditions.

We then compared the wind speed at the two probable regions of emission, Gulf of Lion and Bay of Biscay, to the concentration of PMA, using FLEXPART results, for the PMA period (22-26 June). To account for the transport time of PMA, we added a delay of 12 hours that corresponds to the mean transport time from the Gulf of Lion to Ersa modelled with FLEXPART for the PMA period and 60h for the Bay of Biscay (Fig. 8b and Fig. 8c respectively). This work was done for the PMA period, from 22-26 June. In Fig. 8a, the correlation between the mass concentration of PMA and the wind speed at Ersa is good for fresh PMA (r$^2$= 0.71) (Fig. 8a) as presented in the previous paragraph for the whole campaign. For the Gulf of Lion (Fig. 8 b)), the correlation is good for aged PMA (red curve, r$^2$= 0.87) while there is no correlation following this fit for fresh PMA and wind speed at the Gulf of Lion. The same analysis was done for the Bay of Biscay (Fig. 8c) but no correlation was found for fresh , aged PMA or all the PMA regardless of their ageing.

According to these results, during the PMA period, the PMA that were measured in Ersa were a mixture of fresh PMA emitted near the Ersa station and of aged PMA emitted from the Gulf of Lion. It should be noted that measurements of PMA have also been made when the air masses were coming from the East, but the concentrations were lower ($< 2$ $\mu$g m$^{-3}$).

From these results, the most probable zone that brings PMA to Ersa during ADRIMED regarding altitude, transport time of air masses and local wind speed would be the Gulf of Lion and the sea close to Ersa, considering that the buoy at the Bay of Biscay represents the wind speed of the area. Beyond the scope of this work, an analysis of the emission and transport of marine aerosols during this PMA period is ongoing, using the Méso-NH model.

### 3.2.3 PMA physical properties

In this section, the number and volume size distribution of PMA are investigated, as they are fundamental parameters to estimate the aerosol radiative effects.

Before conducting comparisons on sea salt physical properties, the PMA period was divided into several shorter periods according to their ageing (see Sect. 3.2.1), that will be called 'ageing periods'. In addition, we chose a supplementary period (1-4 July) corresponding to low PMA concentration when it does not exceed the background concentration ($0.76$ $\mu$g m$^{-3}$). The





number and volume size distribution were averaged over the ageing periods and fitted under the assumption that the distribution is a sum of lognormal modes to investigate if the ageing of PMA could be characterized by their size distributions. Three to six modes were necessary to fit the observed dry size distributions.

For the number size distribution, a large variety of Aitken and accumulation mode can be derived when comparing the different periods. They show a large variety of diameters and concentration whether they contain PMA or not, aged or fresh. However, a coarse mode (modal diameter of 1.2 $\mu$m) appears for all the size distributions containing PMA, for both aged and fresh aerosols. This mode does not exist when the concentration of PMA is within the background. The concentration of this mode seems to be higher for fresh than aged PMA which is probably due to dry deposition during transport. As we did not find any significant difference between the size distribution of aged and fresh PMA, they are merged for the sequence of the analysis as PMA size distribution over PMA period.

The number and volume size distribution have been averaged for each periods; Dust, PMA and BBP (Fig. 10). We chose to average the most intense part of each period to extract the representative properties of each aerosol type. Thus, although the dust event starts on 16 to end on 20, we analysed the size distribution obtained from 17 00 UTC to 19 June 24 UTC. Likewise, the study of the size distribution for PMA and BBP aerosols are from 23 June 00 UTC to 25 June 24 UTC and from 9 July 00 UTC to 11 July 24 UTC, respectively. The results are summarized in Table 3. The highest number concentration for PMA period was for particles of modal diameter of 40 nm, followed by a mode at 130 nm and a third mode at 1.2 $\mu$m. We find a good agreement of modal diameters with the size distribution measured by Ovadnevaite et al. (2014) in the parametrisation of the emission of PMA from the Atlantic Ocean. Furthermore, our results agree with measurements performed in the Mediterranean Sea by Schwier et al. (2015). We observed a high number concentration of fine particles during PMA period, which is consistent with measurements reported in Fig. 10 a. The modal diameter of these fine particles is situated at 40 nm. This mode was also measured by Schwier et al. (2015) at d=37.5 nm $\pm$ 1.4 during PMA flux measurements from Mediterranean waters. The second mode has a modal diameter of 130 nm, which is somewhat higher than the 90 nm mode found by Schwier et al. (2015), which is related to the presence of aged particles during our study.

As expected, the coarse mode dominates the volume concentration (Fig. 10b), with a modal diameter of 1.6 $\mu$m. Marine aerosols whose diameters are greater than 1 $\mu$m are largely inorganic sea salt (O'Dowd et al., 2004). Furthermore, the volume size distribution retrieved from AERONET measurement (Fig. 11) is also higher for the coarse mode comparatively to the fine mode. AERONET data are used as a comparison to the in-situ measurements, as they are derived from an algorithm, averaged over a few days period, and with a limited number of measurements (7 available for the PMA period).

### 3.2.4 PMA optical properties and local shortwave direct radiative forcing

As we have highlighted, the PMA period is mainly influenced by marine aerosols, we can obtain the atmospheric optical conditions and local SW direct radiative forcing corresponding to what we define as background conditions. In addition to chemical and size distribution aerosol properties, we also determined optical properties providing AOD at the measuring site, as well as the SSA and AE obtained for the whole atmospheric column from AERONET/PHOTONS observations (Dubovik et al., 2002b) and their spectral dependences in the solar spectral region. These results are summarized in Table 4.



First, the AOD retrievals provide information about the loading of aerosols within the atmospheric column. During the ChArMEx-ADRIMED campaign, AOD (at 500 nm) was found to be moderate, with an average of $0.15 \pm 0.08$ (Fig. 12 a). Such values are consistent with the site location and aerosol concentration (see part 3.3.1), Ersa being not impacted by local pollution or high anthropogenic sources. In that sense, the AOD background is low, typical of a rural site. AOD is lowest during

PMA event (22-26 June), with a mean value of $0.11 \pm 0.08$ at 500 nm, close to those reported over the Mediterranean basin (Smirnov et al., 2002; Pace et al., 2006; Fotiadi et al., 2006).Smirnov et al. (1995) found that for clean oceanic conditions, AOD was below 0.1 (at 550 nm) and Pace et al. (2006) found an average value of 0.11 for the same marine conditions. In the Mediterranean sea in particular, Fotiadi et al. (2006) reported a value of 0.15 in Crete for background situation corresponding to marine aerosols and Mishra et al. (2014) found a mean AOD over the Eastern Mediterranean for marine aerosols (June-August

2010) of $0.06 \pm 0.01$. Furthermore, it should be noted that AOD is not very sensitive to the wavelengths during these 5 days, due to the presence of coarse particles.

For PMA episode, AE varied between 0.4 and 2, with a mean value of 1.3, which is below the mean value of the ADRIMED campaign (1.8) (Fig. 12 b). AE also decreased to 1.15 for 24 June, when PMA concentration is the highest. Such a value is characteristic of clean ocean regions as reported by Smirnov et al. (2002), who found values between 0.3 and 0.7. In addition,

Pace et al. (2006) and Fotiadi et al. (2006) reported AE comprised between 0.7 and 1 for background marine atmosphere over the Central and Eastern Mediterranean. The AE measured at the Ersa station during PMA event is not as low as these referenced values and could indicate a possible mixing between sea-salt and other aerosols, as the Western Mediterranean is under the permanent influence of continental sources. This point is also consistent with the observed number size distribution, which showed that the number concentration of fine particles was high during PMA event, indicating pollution particles from

European continent.

During PMA period, SSA was found close to unity (mean of $0.98 \pm 0.02$) (not shown here), indicating significant scattering optical properties, consistent with marine aerosols optical properties in the solar range (Lewis and Schwartz, 2004).

In addition to the atmospheric column informations, the scattering coefficient measured at the ground by the nephelometer was found to be low (mean of $22 \pm 8$ Mm$^{-1}$) during PMA period, and not sensitive to the wavelengths (Fig. 12 c). The optical

characteristics (AOD, SSA and AE) of the air masses during the PMA event are found to be consistent with the literature (Smirnov et al., 2002; Pace et al., 2006), even though a mixing with continental fine particles was also detected.

In parallel to optical properties observations, the local 1-D (clear-sky) direct radiative forcing (DRF) in the short wave (SW) spectral region has been estimated using AERONET/PHOTONS retrievals (García et al., 2008) for each identified period. DRF is calculated here at two different atmospheric levels, at the surface (or bottom of the atmosphere, BOA) and at the top of the

atmosphere (TOA). Figure 13b indicates the SW DRF at BOA for different AOD and different solar angles observed during the experiment. The estimated values show a significant variability with instantaneous DRF comprised between -5 to -40 $Wm^{-2}$, depending on the aerosol regimes. Figure 13 a and b) indicates that PMA period is characterized by moderate TOA DRF (mean of $-8 \pm 3$ W m$^{-2}$) and BOA DRF (mean of $-11 \pm 4$ W m$^{-2}$). Such estimates at Ersa station are found to be consistent with sea salt direct SW forcings documented by Lundgren et al. (2013) using COSMO-ART model over the Mediterranean basin, who

reported a SW DRF from -5 to -10 W m$^{-2}$ at the surface and for an AOD comprised between 0.1 and 0.2 (at 550 nm).





### 3.3 Comparison of primary marine aerosols properties with dust and BBP events

This PMA period represents the background atmospheric conditions that affect Ersa most of the time. In this section, we compare its physical, optical and direct radiative forcing properties to two sporadic events (Dust and BBP) that influence Ersa principally in spring and summer.

#### 3.3.1 Physical properties

The total number concentration (CPC + OPS) during the campaign observe a mean value of $1900 \pm 920 \, \mathrm{cm^{-3}}$ with several short episodes (few hours) of high concentrations ($> 5000 \, \mathrm{cm^{-3}}$) at the end of June. Thus, the background number concentration is higher than what is usually measured in a marine pristine site ($300\text{-}600 \, \mathrm{cm^{-3}}$) (D O'Dowd and De Leeuw, 2007) and denotes a contamination by other sources, principally from continental Europe, as Ersa is not affected by immediately local sources. In parallel, the number size distributions measured by the SMPS show that the particles detected during these short episodes of high concentration have diameters below 50 nm and probably corresponds to new particles during transport over the Mediterranean sea.

During this field campaign, the fine and accumulation modes ($10 \, \mathrm{nm} < Dp < 600 \, \mathrm{nm}$) were dominant in number. Furthermore, the concentration of these two modes rises in the beginning of July, in particular the accumulation mode, following the scheme already mentioned in the previous section for $PM_1$ particles. Hence, the ratio of the number concentration from 4-13 July over the number concentration from 6 June to 3 July is greater than 2 for particle diameters greater than $0.24 \, \mu$m and $0.52 \, \mu$m. This ratio reaches its highest value for particles diameter of $0.4 \, \mu$m.

Concerning the number size distribution, we find important distinctions between the three different periods as reported in Fig. 10a. As expected, during the dust event, the number size distribution is higher for the largest particles (3 to 10 $\mu$m size range). During BBP period, the dominant mode of the number size distribution is located around 200 nm and the number concentration of particles greater than 500 nm is found to be low ($65 \pm 14 \, \mathrm{cm^{-3}}$). This result is consistent with the typical number concentration of biomass burning aerosols that peaks in the size range of 100-200 nm (Guyon et al., 2005; Reid et al., 2005; Andreae et al., 2007). As these hydrophilic aerosols are subject to increases in size when they age during transport (Andreae and Rosenfeld, 2008), this is consistent with our observations of a mode centred at 200 nm as they were transported for 3-4 days before reaching Ersa.

Looking at the volume size distribution is a way to distinguish the particles that have the greatest impact on the mass concentration, i.e. the coarser particles. On average, during the ADRIMED period the mean total volume concentration (CPC + OPS) is $40 \pm 16 \, \mu\mathrm{m^3 \, cm^{-3}}$, and the volume concentration of smallest particles (d<500 nm) is $22 \pm 11 \, \mu\mathrm{m^3 \, cm^{-3}}$ while the coarser particles (d>500 nm) is $16 \pm 9 \, \mu\mathrm{m^3 \, cm^{-3}}$.

The volume size distribution shows different patterns for the Dust, PMA and BBP periods. We distinguish a coarser mode between 20-27 June, including PMA period. A coarse mode is also observed around 19 June (Dust period) with diameters between 5-7 $\mu$m, that probably corresponds to mineral dust particles in accordance with the volume size distributions measured on-board the ATR-42 aircraft (Denjean et al., 2016).





Figure 10b shows two dominant modes during the dust period; one at a dry diameter of 0.18 $\mu$m and the second one around 2.4 $\mu$m. During the PMA period, Fig. 10b indicates a dominant mode situated in the coarse part of the aerosol size distribution, at 1.64 $\mu$m. Finally, the BBP event is found to be dominated by a mode at 320 nm, and the volume concentration of the coarse mode is here very low.

We have also compared the results of the in-situ surface volume size distributions with AERONET/PHOTONS retrievals (Fig. 11). Concerning Dust and PMA periods, the coarse modes measured by OPS and SMPS are consistent with the atmospheric column volume size distribution and contributes to the largest fraction of aerosol mass. During the BBP period, both observations (in-situ and AERONET) clearly indicate volume size distributions largely dominated by the fine mode.

These three periods are characterized by different volume size distribution (in-situ measurements), as summarized in Table 3. The Dust and PMA periods are characterized by coarser particles, with a modal diameter of 2.4 and 1.6 $\mu$m respectively, while the BBP period is characterized by particles in the accumulation mode with modal diameter of 320 nm.

### 3.3.2 Optical properties and local shortwave direct radiative forcing

The aerosol optical properties of Dust and BBP periods are presented here as a comparison with PMA period.

The AOD is low during the first part of the campaign, until the beginning of July, except for two short events where it reaches values as high as 0.4. From the beginning of July to the end of the campaign, the AOD increases to values up to 0.6 with a higher wavelength dependency. During the campaign, the average value of the AOD is moderate ($0.15 \pm 0.08$) but during the dust event, it increases to reach 0.3 (at 500 nm) (Fig. 12), corresponding to a relatively low value for a dust outbreak occurring over the Mediterranean basin (Mallet et al., 2016). AOD can reach values above one (Guerrero-Rascado et al. (2009), over the Western Mediterranean) and even up to two (Di Sarra et al. (2011) over Lampedusa). As observed during PMA period, AOD is not sensitive to wavelengths during the dust event, denoting the presence of coarse particles. AOD showed a very different pattern during the last part of the campaign, reaching higher values and showing a strong dependence to the wavelengths. AOD thus exceeds 0.4 in the middle of July and is higher for shorter wavelengths. It denotes a significant contribution of small particles to the solar extinction, in accordance with SMPS and TEOM PM$_1$ observations previously presented and during the BBP period. The AOD values are much higher for these two periods than during PMA period.

Concerning AE, during the field campaign, we distinguished two main periods; June and July. During the first one, the angstrom exponent is mostly less than two and follows significant and rapid variations, which denotes a high variability in the size and type of aerosols (Fig. 12). On the contrary, during July, AE is above two and we can clearly note a diurnal variation, which is absent in June. During July, the mass concentration of PM$_1$ particles was almost twice as high as those in June, and the higher concentrations of small particles (diameter $< 1$ $\mu$m) is consistent with the observed increase of AE. In particular, during the dust episode AE fluctuated between 1-2, which are not typical values observed for desert dust particles, which generally tends toward values less than one, denoting a majority of coarse particles (Dubovik et al., 2002a). In that sense, the higher values observed at Ersa could be due to the possible mixing of particles in the atmosphere during these days, by the weak intensity of the dust outbreak observed during ADRIMED or by the possible deposition of the coarser dust particles during transport. Finally and during BBP period, AE was found to be mostly above two. Its pattern follows a clear diurnal variation,



with a maximum around 12 UTC and a minimum in the beginning and in the end of the day. AE observed during this period is stable for almost a week, from 4-10 July. The largest difference noted for AE between Dust, PMA and BBP periods is in their internal variability. For the first two periods, a mixing and high variability is found while for the last period, AE is constant for more than 5 days, showing that the atmosphere is mostly under the influence of the same aerosol type.

Overall, SSA observed during the campaign remained relatively high, with values above 0.90 for most of the period, associated to a spectral dependence less than 0.05 (from 440 to 870 nm). In that sense, the presence of absorbing particles is shown to be sporadic and lasted no more than a few hours. However, for some short episodes, the value decreased below 0.9 and even 0.7. This corresponds to smaller particles and specifically to biomass burning or polluted aerosols (15 and 19 June and 1, 2, 6 and 9 July) (Dubovik and King, 2000). During Dust period, 16-20 June, SSA decreased to values between 0.90

to 0.95 (at 440 nm), indicating moderate absorbing properties, which are characteristics of desert dust over the Mediterranean basin (Mallet et al., 2013). Finally and during BBP period, we observed a higher wavelengths dependency, with SSA values oscillating between 0.90 and 1.0 (at 440 nm).

Over the entire period of the campaign, nephelometer measurements reveal that the scattering due to particles was relatively low (mean of $29\,\mathrm{Mm}^{-1} \pm 16$, at 550 nm) and not sensitive to wavelength during June and in particular during Dust and PMA

periods. This is in contrast to July, where higher scattering coefficients (mean of $37\,\mathrm{Mm}^{-1} \pm 18$) associated with higher AE (AE July mean of $2.1 \pm 0.2$, AE June mean of $1.6 \pm 0.5$) are observed. During the campaign, the scattering coefficient at the ground measured at 550 nm varied between 0 and $50\,\mathrm{Mm}^{-1}$ for June and the beginning of July, and reached higher values up to $100\,\mathrm{Mm}^{-1}$ from 8 to 14 July, as shown in Fig. 12 c). The averaged scattering coefficient is $24 \pm 12\,\mathrm{Mm}^{-1}$ in June, twice lower than observed during July (mean value of $49\,\mathrm{Mm}^{-1}$ at 550 nm). In July, the measured scattering coefficient is very sensitive to

the wavelengths and the mean difference between the scattering coefficient estimated in the blue and red wavelengths is about $44 \pm 13\,\mathrm{Mm}^{-1}$ from 8 to 14 July (compared to $17 \pm 9\,\mathrm{Mm}^{-1}$ in June). This clearly indicates that aerosols are smaller in size during this period, which is well consistent with AERONET/PHOTONS data and $PM_1$ concentrations obtained at Ersa station. During Dust and PMA period, the scattering coefficient remains low (mean of $21 \pm 9\,\mathrm{Mm}^{-1}$ and $22 \pm 9\,\mathrm{Mm}^{-1}$ respectively), without wavelength dependency. On the contrary, during BBP period, the wavelenght dependency is the highest (mean of 37

$\pm 11\,\mathrm{Mm}^{-1}$ between 450 nm and 700 nm), and the scattering coefficient reaches highest values (up to $106\,\mathrm{Mm}^{-1}$).

Finally, the study of the local radiative forcing shows that the highest values of BOA DRF and TOA DRF correspond to highest AOD observed during dust event. For this specific event, values peak maxima of $-43\,\mathrm{W\,m}^{-2}$, that are in the same range of magnitude of values reported for mineral dust aerosols over the Mediterranean basin by Di Biagio et al. (2010). Intermediate BOA DRF are calculated under polluted and biomass burning influence (from 5-12 July) with DRF ranging from -13 to -38 W

$\mathrm{m}^{-2}$. Such values are classically derived over the Western Mediterranean for polluted particles (Roger et al., 2006).

In addition, the calculated SW DRF at TOA is reported in Fig. 13b, showing negative forcings in all situations, due to the moderate absorbing ability of aerosols associated to a low surface albedo at Ersa (Nicolas et al., in prep.), leading to a cooling at TOA. It should be noted that DRF of aerosols in the long wave (LW) spectral range, which can counterbalance a part of the SW cooling at TOA, is not estimated here. Contrary to the LW DRF of mineral dust exerted near dust sources, this effect is

generally lower than SW DRF during the transport of mineral dust over the Mediterranean basin (Nabat et al., 2015). In the





same way as at the surface, Fig. 13 b) indicates that higher TOA DRF occur during the mineral dust event, with values as large as -20 to -25 W m$^{-2}$, but due to the spread of the values during the episode, the mean value of TOA DRF is in the same range than for BBP period. Finally, we report logically intermediate TOA DRF (mean of -15 $\pm$ 4 W m$^{-2}$) between 5 to 12 July, when Ersa station is affected by pollution and smoke aerosols.

To conclude, PMA SW DRF at TOA and BOA is 2 or 3 times lower than what we encounter during events like dust outbreaks and biomass burning, which occur principally in spring and summer. However, the influence of marine aerosols is permanent, depending particularly on wind speed.

## 4   Conclusion

The ChArMEx-ADRIMED campaign that took place in summer 2013 in the Western and Central Mediterranean basins has
served to characterize the aerosol chemical, physical and optical properties, to quantify their direct radiative forcing and study their implications on the regional climate (Mallet et al., 2016). One of the ground-based instrumented sites was based in Ersa, Cap Corsica and allowed the study of different aerosol types, particularly the properties and relative impacts of PMA compared to other aerosol types present in the western Mediterranean basin.

    Using FLEXPART back-trajectory simulations and in-situ chemical, physical and optical measurements, we show that Ersa
was impacted by air masses coming from different source regions and bringing different aerosol types. Three main periods have been identified, to characterize the relative impacts of the major aerosol types and in particular, a period (22-26 June) when the Ersa site is mainly affected by PMA. During this period, the Ersa station was influenced by westerly wind, bringing air masses from Gulf of Lion, Mediterranean coasts of France and Spain and the Bay of Biscay. During this specific event, the concentration of PMA was relatively high, reaching 6.5 $\mu$g m$^{-3}$, which represents 40% of the total PM$_{10}$ mass concentration.
Here, an original dataset, obtained from ATOFMS and PILS-IC instruments has been used to study the ageing of PMA. By comparing the two instruments, we found that the majority of the time PMA had already undergone chemical reactions and so were not freshly emitted near Cap Corsica, but rather advected from long-range transport. In particular, during the PMA period, based on FLEXPART simulations and local wind speed measurements, we distinguished the origin of fresh and aged PMA composing the mixing of PMA observed in the Ersa station; we found that fresh PMA were emitted near the station
under high wind speed conditions while aged PMA were most probably originating from the Gulf of Lion (Mediterranean) and not from the Bay of Biscay (North Atlantic Ocean).

    These two original instruments display similar results regarding PMA ageing, and detect different short periods (of few hours duration) of mostly aged or mostly fresh PMA dominance that we used for our analysis.

    No significant distinction was found between the number size distribution of fresh and aged supermicron PMA, and so the
size distribution was fitted regardless of the ageing of these aerosols. The lognormal modes (4 modes with diameters of 0.04, 0.13, 1.2 and 5.4 $\mu$m) found for these PMA were in agreement with previous measurement made by Ovadnevaite et al. (2014) and Schwier et al. (2015). The PMA episode was also influenced by fine particles, denoted by the high number concentration of fine particle and by an Angstrom Exponent varying between 0.4 to 2. In parallel, low AOD (mean of 0.11 at 500 nm) and




SSA (at 440 nm) close to unity also measured at Ersa are typical of the PMA influence. The SW DRF showed the lowest values at the surface compared to other aerosol regimes with a mean of -11 $\pm$ 4 W m$^{-2}$. At the top of the atmosphere (TOA), the lowest values were also observed during PMA event (mean of -8 $\pm$ 3 W m$^{-2}$).

The aerosol properties obtained during this PMA event were compared to two other periods encountered during the field campaign (Dust and BBP). The first period corresponds to a dust outbreak of moderate intensity (16-20 June; Dust), and the last period (5-12 July) is characterized by biomass burning that originated in Ukraine mixed with pollution for Southern Europe (BBP). In terms of physical and chemical properties, our results display large variability in the number and volume size distribution as well as mass concentrations between the different events. The volume size distribution analyses reveal that the BBP event is dominated by a fine mode of particles with a modal diameter of 320 nm, while the PMA period is dominated by a coarse mode with a modal diameter of 1.64 $\mu$m. Finally, dust aerosols observed at Ersa are characterized by a modal diameter of 2.4 $\mu$m, which is found to be consistent with aircraft in-situ observations within dust plumes in the free troposphere during the airborne portion of ChArMEx-ADRIMED experiment (Denjean et al., 2016).

Concerning the optical properties, our results indicate that the dust event is characterized by a moderate AOD with a mean of 0.16 $\pm$ 0.08 and highest values reaching 0.30 (at 500 nm) associated to a mean AE of 1.4 (calculated between 440 and 870 nm). SSA during the dust episode (0.97 at 440 nm) is found to be high, revealing mostly scattering dust aerosols in this case.

The most intense optical signature occurs clearly at the end of the campaign during the BBP episode. For this specific period, a significant scattering coefficient estimated at the surface (mean of 42 Mm$^{-1}$ at 550 nm, together with moderate AOD (0.23 $\pm$ 0.07 at 500 nm) and elevated spectral dependence are observed.

In terms of SW DRF, our results showed the highest (lowest) contribution to surface forcings during the dust (PMA) event with -21 $\pm$ 11 W m$^{-2}$ (-11 $\pm$ 4 W m$^{-2}$), with intermediate values (-23 $\pm$ 6 W m$^{-2}$) observed during the BBP episode. All derived SW DRF at the surface for the three aerosol types are similar to previous studies in the Western Mediterranean Basin. Similar results are obtained for the top of the atmosphere (TOA) forcing, with the highest values occurring during the dust outbreak (-14 $\pm$ 6 W m$^{-2}$) and BBP period (-15 $\pm$ 4 W m$^{-2}$), while the lowest values were observed during the PMA event (-8 $\pm$ 3 W m$^{-2}$). Even though the magnitude of PMA DRF is relatively small compared to Dust and BBP DRF, its impact is permanent due to the persistency of PMA in the marine atmosphere.

*Author contributions.* PILS-IC, ACSM, TEOM PM10, TEOM PM1 data were provided by J. Sciare; ATOFMS data were provided by J. Arndt and J. Wenger; SMPS, OPS, CPC, Nephelometer, meteorological data were provided by Météo-France. M. Claeys prepared the manuscript with contributions from all co-authors.

*Acknowledgements.* This research was supported by Direction Générale de l'Armement (DGA) and Météo-France. This research has received funding from the French National Research Agency (ANR) project ADRIMED (contract ANR-11-BS56-0006). This work is part of the ChArMEx project supported by ADEME, CEA, CNRS-INSU and Météo-France through the multidisciplinary programme MISTRALS (Mediterranean Integrated Studies aT Regional And Local Scales). The station at Ersa was partly supported by the CORSiCA project funded by the Collectivité Territoriale de Corse through the Fonds Européen de Développement Régional of the European Operational Program




2007-2013 and the Contrat de Plan Etat-Région. We acknowledge the AERONET/PHOTONS sun-photometer networks and the PI of the Ersa station and their staff for their work to produce the data set used in this study. Contributions by Thierry Bourrianne are gratefully acknowledged. Thanks to Laurent Gomez without whom this study would have never taken place.





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





**Table 1.** Transport time (Mean ± standard deviation) from Gulf of Lion and North Atlantic Ocean to the Ersa station obtained from Flexpart simulations analyses (see Sect. 3.1.1)

|  | 22 June | 23 June | 24 June | 25 June | 26 June |
|---|---|---|---|---|---|
| Gulf of Lion : Mean Transport time | | | | | |
| Days | 0.72 ± 0.69 | 0.49 ± 0.29 | 0.21 ± 0.09 | 0.55 ± 0.30 | 1.32 ± 0.50 |
| North Atlantic Ocean : Mean Transport time | | | | | |
| Days | 2.88 ± 0.87 | 2.50 ± 0.72 | 2.94 ± 1.5 | 3.23 ± 1.26 | 4.45 ± 1.22 |





**Table 2.** Caracteristics of the three log-normal modes of aged sea salt aerosols measured by ATOFMS at Ersa station

| Mode | Aerodynamical diameter | $\sigma$ |
|:---:|:---:|:---:|
| | $\mu$m | |
| 1 | 0.46 | 1.28 |
| 2 | 1.13 | 1.35 |
| 3 | 1.95 | 1.23 |





**Table 3.** Characteristics of the fit by a lognormal distribution (N, d, $\sigma$) for the three periods : Dust, PMA and BBP

| Number Concentration | $N_1$ | $d_1$ | $\sigma_1$ | $N_2$ | $d_2$ | $\sigma_2$ | $N_3$ | $d_3$ | $\sigma_3$ | $N_4$ | $d_4$ | $\sigma_4$ |
|---|---|---|---|---|---|---|---|---|---|---|---|---|
| Dust | 156 | 0.06 | 1.88 | 389 | 0.13 | 1.51 | 0.11 | 1.16 | 1.3 | 0.02 | 3 | 1.47 |
| PMA | 1162 | 0.04 | 1.46 | 164 | 0.13 | 1.56 | 0.45 | 1.2 | 1.5 | 0.02 | 5.4 | 1.25 |
| BBP | 0.13 | 0.027 | 0.9 | 582 | 0.08 | 1.79 | 170 | 0.22 | 1.35 | 0.07 | 1.5 | 1.7 |
| Volume Concentration | $V_1$ | $d_1$ | $\sigma_1$ | $V_2$ | $d_2$ | $\sigma_2$ | $V_3$ | $d_3$ | $\sigma_3$ | $V_4$ | $d_4$ | $\sigma_4$ |
| Dust | 0.5 | 0.18 | 1.46 | 0.64 | 0.26 | 1.43 | 0.43 | 2.36 | 1.62 | | | |
| PMA | 0.09 | 0.07 | 1.47 | 0.44 | 0.24 | 1.54 | 1.02 | 1.64 | 1.71 | 0.18 | 6.66 | 1.34 |
| BBP | 0.77 | 0.2 | 1.54 | 1.34 | 0.32 | 1.33 | 0.31 | 2.24 | 1.55 | 0.2 | 6.13 | 1.36 |





**Table 4.** Summary of the optical properties (mean and standard deviation) estimated for the three different aerosols regimes: AOD, AE, SSA, Scattering coefficient (in $\mathrm{Mm}^{-1}$), and instantaneous TOA and BOA radiative forcing (in $\mathrm{W\,m}^{-2}$)

|  | AOD | AE | SSA | Scattering coefficient | TOA | BOA |
|---|---|---|---|---|---|---|
|  | 500 nm | 440-870 nm | 440 nm | 550 nm |  |  |
| Dust | $0.16 \pm 0.08$ | $1.4 \pm 0.3$ | $0.97 \pm 0.03$ | $21 \pm 9$ | $-14 \pm 6$ | $-21 \pm 11$ |
| PMA | $0.11 \pm 0.08$ | $1.3 \pm 0.4$ | $0.98 \pm 0.02$ | $22 \pm 9$ | $-8 \pm 3$ | $-11 \pm 4$ |
| BBP | $0.23 \pm 0.07$ | $2.1 \pm 0.2$ | $0.98 \pm 0.03$ | $42 \pm 14$ | $-15 \pm 4$ | $-23 \pm 6$ |





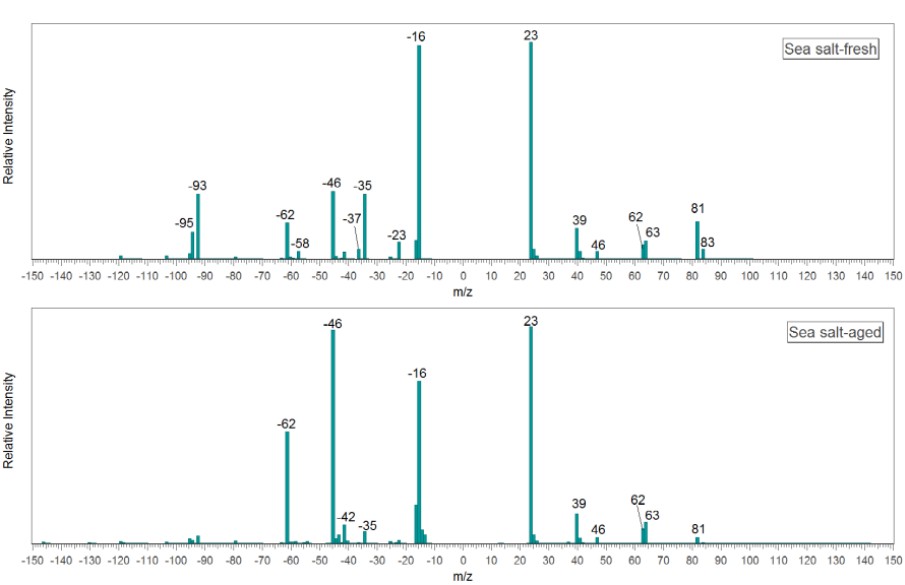

**Figure 1.** Average mass spectra for fresh and aged sea salt particles observed during ADRIMED.



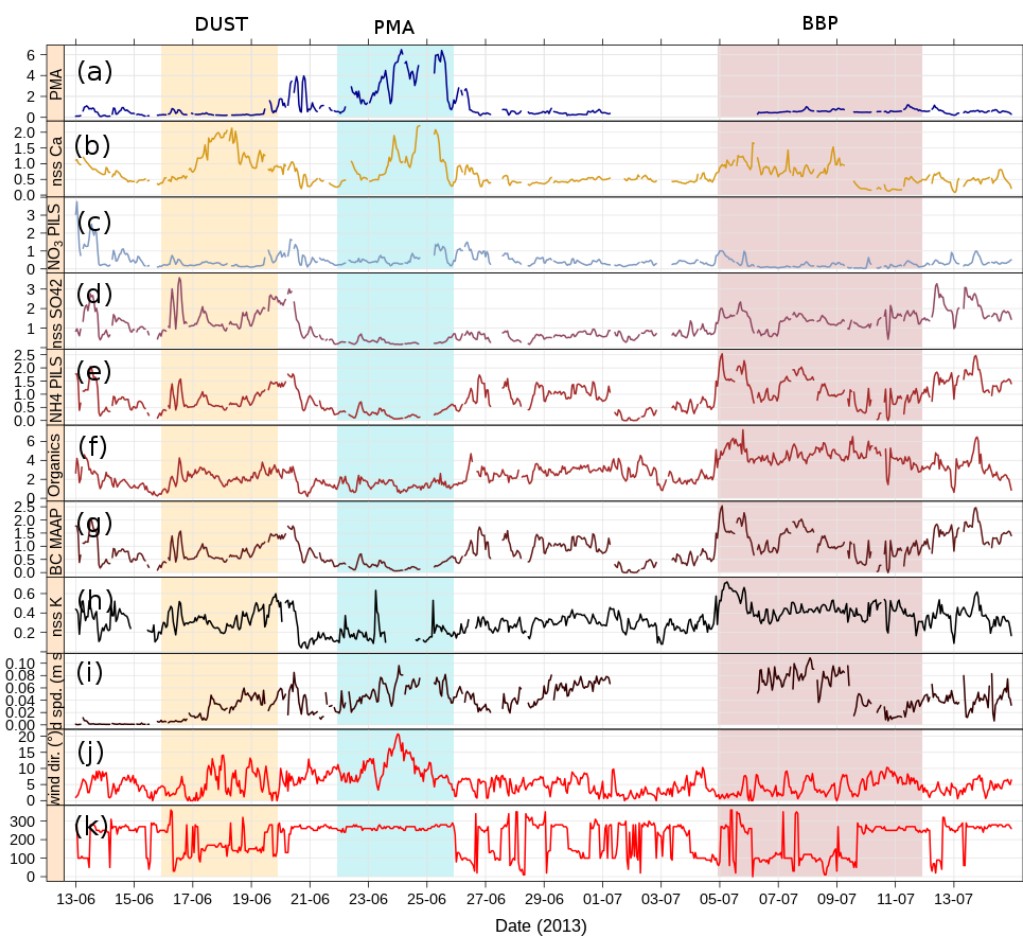

**Figure 2.** Time series of chemical species (in $\mu g\ m^{-3}$), wind speed (m s$^{-1}$) and direction (°): (a) PMA mass concentration calculated from PILS-IC measurements , (b) nss-Ca$^{2+}$ mass concentration measured by PILS-IC measurements, (c) nss-NO$_3^-$ mass concentration measured by the PILS-IC, (d) nss-SO$_4^{2-}$ mass concentration measured by the PILS-IC, (e) nss-NH$_4^+$ mass concentration measured by the PILS-IC, (f) Organic mass concentration measured by the ACSM, (g) Black carbon mass concentration measured by the MAAP, (h) nss-K$^+$ mass concentration measured by the PILS-IC, (i) Wind speed measured at the Semaphore, (j) Wind direction measured at the Semaphore. The shaded aeras correspond to the three identified periods, orange for Dust, blue for primary marine aerosols (PMA) and brown for Biomass burning/pollution (BBP).





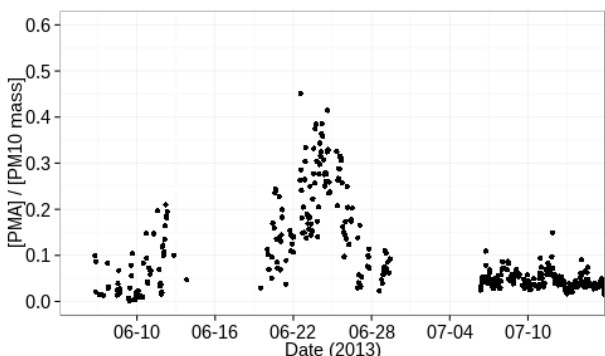

**Figure 3.** Ratio of inorganic sea salt mass concentration (PILS-IC) over $PM_{10}$ mass concentration (TEOM $PM_{10}$).

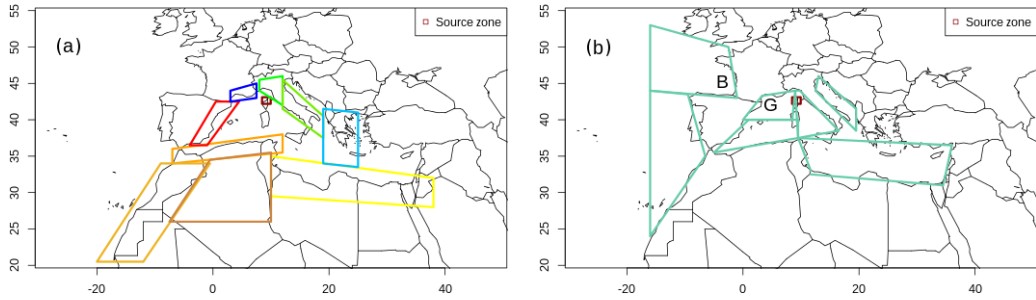

**Figure 4.** Maps representing the different zones used for the study of the origin of air masses with Flexpart. (a) Anthropogenic and desert zones (Red for Spanish coasts, Dark Blue for French coasts, Green for Italy, Blue to Greece, Orange and Yellow for Northern Africa; (b) Marine zones (G corresponds to Gulf of Lion, B to Bay of Biscay).





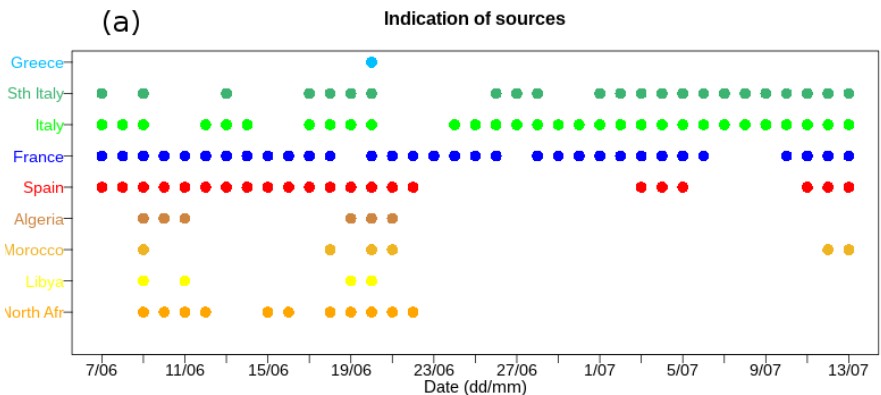

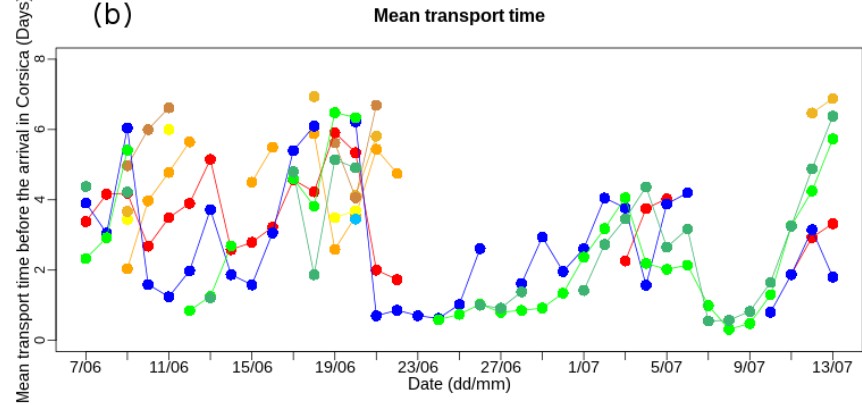

**Figure 5.** Time series of air mass sources derived from the Flexpart back trajectory simulations at 500 m from 07/06/2013 to 13/07/2013. The top figure (a) represents the passage of an air mass through the different zones before they reached Ersa. (b) represents the transport time of the air masses from each zone in (a) to Ersa.





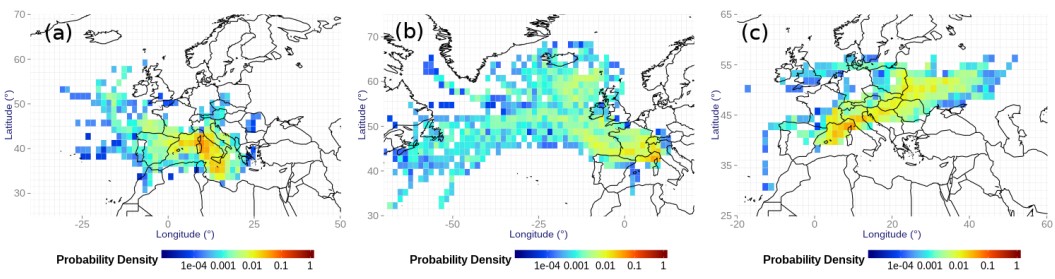

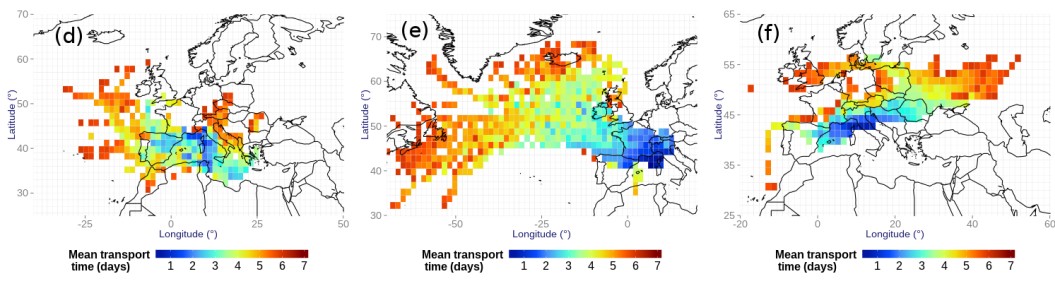

**Figure 6.** FLEXPART maps representing for the top ones ((a), (b) and (c)) the probability density of the back trajectories for the three periods : Dust, PMA and BBP respectively. The bottom ones ((d), (e) and (f)) represents the mean transport time from the Ersa station for the three periods: Dust, PMA and BBP respectively. Each backtrajectory starts at 500 m from the Ersa measurement site.





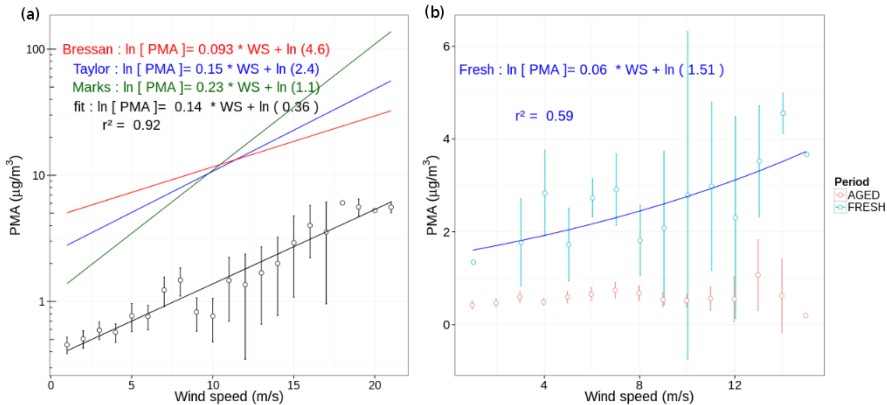

**Figure 7.** PMA concentration measured at Ersa as a function of wind speed for the ADRIMED period. The PMA concentrations have been averaged by wind speed bins of 1 m s$^{-1}$. The errorbars represent $\pm 2\sigma / \sqrt{N}$ (N is the number of independent measurements). (a): the black curve correspond to measurements, while the red, blue and green curves correspond to fits parameters by Bressan and Lepple (1985); Taylor and Wu (1992); Marks (1990). (b): represents fresh (blue curve) and aged (red curve) PMA during the whole campaign. .

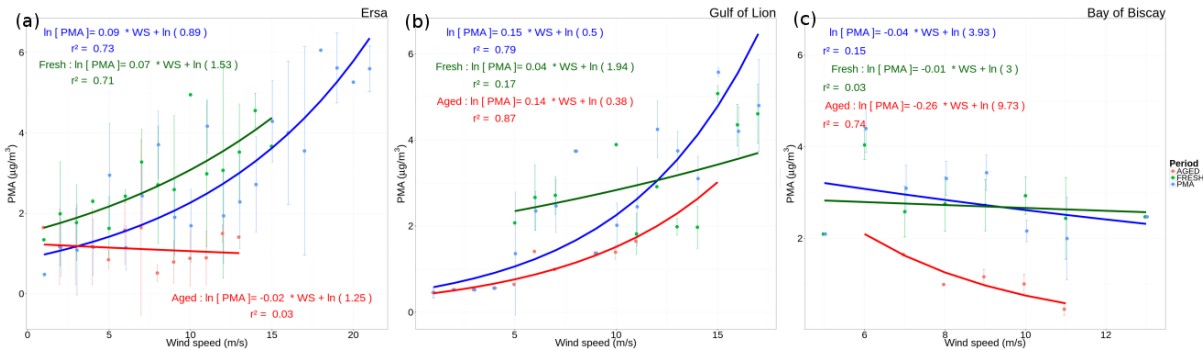

**Figure 8.** Concentration of sea salt aerosols measured by the PILS IC as a function of wind speed measured at Ersa (a), at the Gulf of Lion buoy (b) and at the Bay of biscay buoy (c), for the PMA period. An offset of 12h and 60h has been applied between the wind speed measurements in the Gulf of Lion and the Bay of Biscay respectively and the PMA concentrations observed at Ersa to account for transport time of the air masses. The PMA concentrations have been averaged by wind speed bins of 1 m s$^{-1}$. The error bars represent $\pm 2\sigma / \sqrt{N}$. Blue curves represents all the PMA measurements, while green and red curves represent fresh and aged PMA respectively.





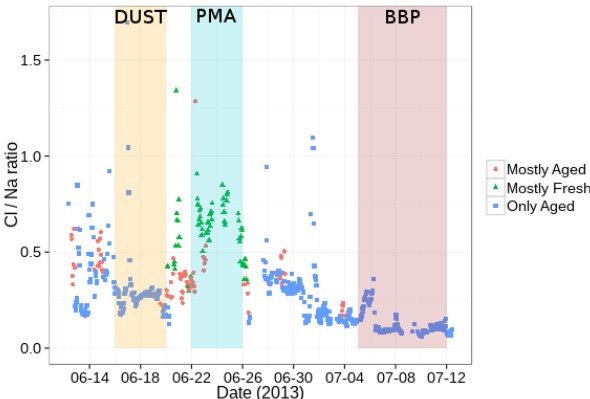

**Figure 9.** Comparison of the two instruments ATOFMS and PILS-IC for inorganic component of sea salt aerosols. The time series represents the mass ratio of chloride to sodium ions calculated from the PILS-IC measurements. The marker color represents the degree of ageing determined by the ATOFMS. .

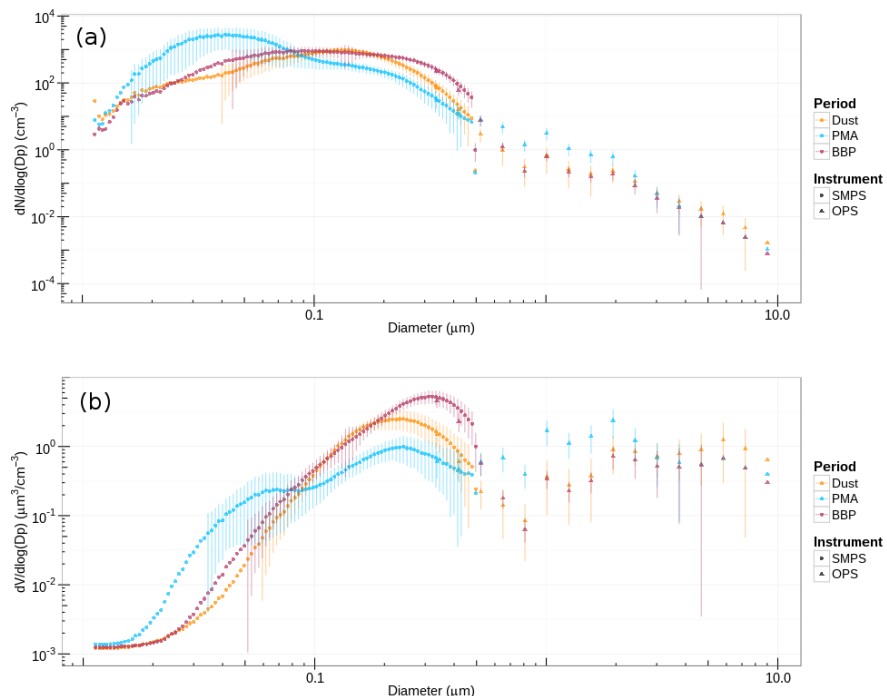

**Figure 10.** Number (a) and volume (b) size distribution averaged by periods of Dust, PMA, and BBP using the SMPS and OPS instruments. The dry diameters range 10 nm to 10 $\mu$m. The first and last day of each period was removed to capture the main feature and the maximum amplitude of the event. .





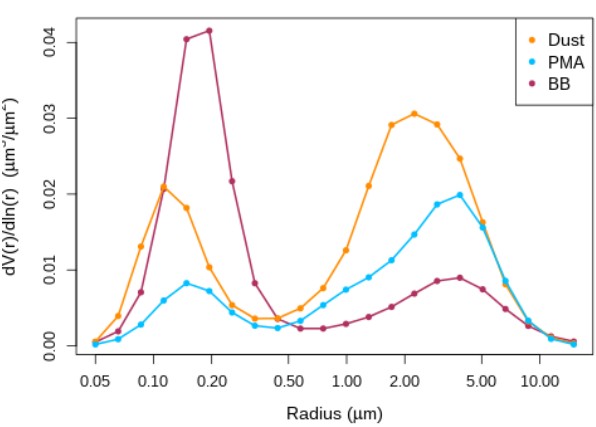

**Figure 11.** AERONET volume size distributions averaged for the three periods .





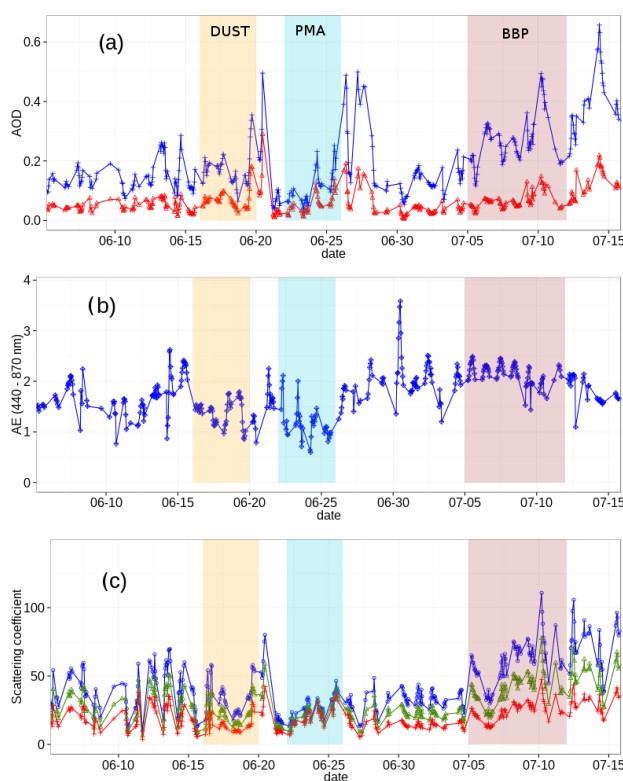

**Figure 12.** Time series of (a) Atmospheric Optical Depth (AOD) at 2 wavelengths ( 440 and 870 nm) measured by the radiometer from the AERONET network situated at the Semaphore during the ADRIMED campaign, (b) Angstrom exponent calculated from AERONET data, during the ADRIMED campaign, using the extinction measurements at 440 and 870 nm, (c) Scattering coefficient at three wavelengths ; 450 nm (blue), 550nm (green) and 700 nm (red) measured by the nephelometer situated at Ersa..





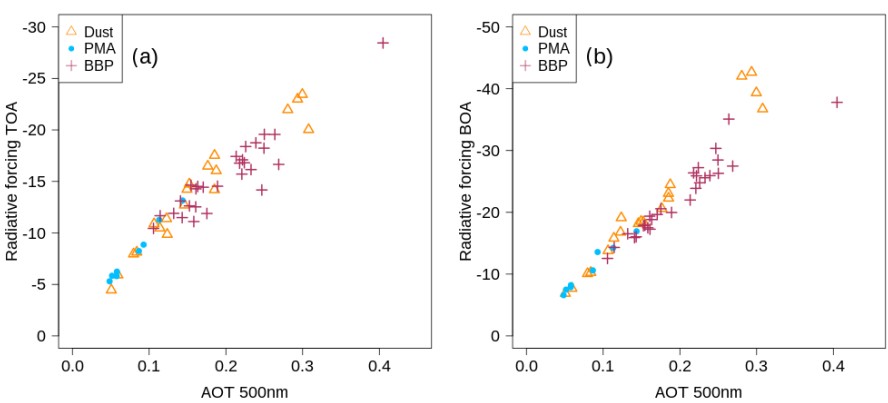

**Figure 13.** Aerosol radiative forcing at (a) the Top of the Atmosphere (TOA) and (b) the Bottom of the Atmosphere (BOA) represented as a function of the aerosol optical thickness (AOT) for each of the major periods.