# Peer review of "Optical, physical and chemical properties of aerosols transported to a coastal site in the Western Mediterranean: Focus on primary marine aerosols"

_Atmospheric Chemistry and Physics, 2016_

## Referee Comment (RC1) · Anonymous Referee #1 · 26 Sep 2016

General comments

The ground-based observations at the Ersa site, as part of the ChArMEx-ADRIMED campaign, were used for characterization of aerosol chemical, physical and optical properties and quantification of the short wave radiative forcing. The study focuses on marine aerosols, providing also the opportunity to compare the aforementioned properties for cases of dust originated and biomass burning transported particulates at the area. The role of aerosol ageing, in conjunction with the prevailing meteorological conditions (air masses origin and velocity), on size distribution and PMA sources was

investigated. A significant variability of chemical composition was encountered for each one of the three periods, whereas the different types of aerosol was depicted on the size and optical/radiative properties. Additionally, the results are well interpreted and cross-referenced. I recommend this manuscript to be accepted for publication after considering the following comments and suggestions.

Specific comments

Despite the fact that the instrumental set up has been described in details, providing also information for calibration and quality control where necessary, it is not clear if the MAAP was sampling through a PM1 (P.5, l. 7) or PM2.5 (P.7, l. 9) inlet. This information could be addressed along with the first reference to MAAP (P.4, l.1-3).

Information about the levels of aerosol (and constituents) or other parameters variability is provided in several sections of the manuscript. The readers would easier read the manuscript if additional or adjusted plots were available. For example, a full description of PM1 and PM10 was provided at P. 7, l. 22-28. The addition of the TEOM PM10 and TEOM PM1 plot as part of Figure 2 or as supplementary material would be substantial. The reconstructed PM10 could also be included, since all these parameters are examined thoroughly in the manuscript (section 3.1). Also a description of AOD at 500 nm is described (P. 15, l. 1-5) related to Figure 12a presenting the AOD temporal variability during the campaign at 440 nm and 870 nm. These two wavelengths are useful for the demonstration of the spectral dependence; nevertheless the authors could consider including the AOD time series at 500 nm also and indicate the different color code in the caption of the figure.

In accordance with the previous comment, the means and standard deviation for each parameter conserning the total period could be added to the summary on Table 4.

A diurnal variability of AE for the second part (July) of the campaign was revealed under the impact of biomass burning (P. 17-18). Nevertheless, an explanation or references of similar variability are not provided. Were the factors controlling the observed diurnal

pattern investigated?

Which is the contribution of nss-ions on the total ionic level overall and for each period independently? Increased nss-ions during dust and biomass burning comparatively to marine influenced period could be additionally used as indication for the presence of other sources at the site apart from marine.

P 14, l. 5: taking into account that the PMA is analyzed in that section it would be more appropriate to comment the low or high marine aerosol concentration instead of the presence or not of marine aerosol. Unless the comment refers to all periods.

Technical corrections

P. 5, l. 27-31: The information about the nephelometer is duplicated. It has already been described in pages 3-4. The comment about the scattering coefficient relation to aerosol size and concentration could be transferred in that point.

Figure 2: I would recommend to authors to check the plots a-k. Please pay attention on the caption and axis labels as well, especially for plots i-k. Namely: Plot i demonstrates very low wind speed. Under my opinion it is not valid. In P.10, l. 3-4 the authors refer that "At the Ersa site, during the dust outbreak, around 19 June, the wind speed reached 15 m s$-1$". Plot j is probably wind speed instead of wind dir (according also to figures 7 and 8, wind speed is up to 20 m s-1). Please indicate what is monitored in plot k. It seems to be wind dir. Furthermore, according to P.8, l. 23-24, BC highest concentration encountered on July 5 was equal to 0.75 $\mu$g m-3. Based on Figure 2g the maximum BC concentration was at the range of 2.5 $\mu$g m-3 (same date) or BC is actually depicted in Figure 2h. Plots e and g seem to be the same.

Typing errors:

P. 13, l.6: in function instead of "in fonction"

P. 18, l.31: SW DRF at TOA is depicted in Figure 13 a, not b.

---

## Referee Comment (RC2) · Anonymous Referee #2 · 4 Oct 2016

At first I want to apologise for the delay of my revision.

The paper by Claeys et al. analyse data on the chemical, physical, and optical properties of aerosols measured at the ground at the Ersa site (Corsica) during Charmex/Adrimed in June-July 2013. Ground-based data are combined with FLEX-PART simulations to estimate the origin of the aerosols and their aging time. Co-located AERONET data are used to estimate the aerosol physical and optical properties over the whole atmospheric column, and their direct shortwave radiative effect. The main results of the paper indicate the occurrence of three aerosol regimes at Ersa

during the investigated period: primary marine aerosols, PMA; African dust; pollution aerosols/biomass burning from Eastern Europe. PMA aerosols represent about 40% of the PM10 mass at the station during the considered period. PMA is composed mostly of aged and almost purely scattering particles. The PMA direct radiative effect estimated at the station from AERONET inversions is lower than that measured for dust and pollution particles.

The aim of the paper, i.e. investigating the role of PMA in the Mediterranean and their properties and radiative effect, is interesting and would deserve publication. However, I have some remarks concerning the data analysis and presentation. In particular, I have two main points regarding the representativeness of surface data to discriminate between different aerosol regimes. The two points are the following:

1/ you do not reach mass closure between TEOM data and PILS/MAAP/ACSM (Section 2.4) and I wonder what is the impact of this on your results. Is the aerosol chemical composition that you find representative of the whole aerosol population, or not? This is a key aspect to validate your results on the aerosol composition and associated aerosol type discrimination.

2/ By looking at Figure 10 I would expect larger differences in the size distribution for the three cases, especially in the coarse part. Instead, size distributions seem to agree within uncertainties for PMA, dust, and pollution/biomass burning aerosols. How can you explain this? For dust, this is due to the fact that, as you say in the paper, particles are mostly located above 3 km, while your measurements are at the ground. On the contrary, when you look at column averaged data (Figure 11), you have very large differences in the size distribution for the three periods. This is reasonable since AERONET data represent atmospheric condition over the whole column. By looking at these two plots, however I wonder how representative surface data are and how well can they be used to correctly discriminate between the three periods. This is a crucial point to validate the results/observations at the surface.

[Figure]

More detailed comments are provided in the following.

Specific comments

Abstract I would suggest the authors to partly rewrite the abstract to put more in evidence the role of marine aerosols, since in the present form it seems to me not fully in line with the title/text of the paper. It seems to me that the accent is put mostly on the estimate of the direct radiative effect of sea salt compared to dust and pollution/biomass burning, while this aspect represent only a part of the paper. I would also suggest adding a sentence at the end of the abstract to highlight your conclusions. Also, but this is a minor thing, throughout the abstract and the paper you use randomly "optical, physical, chemical", "physical, optical, chemical", or "chemical, physical, optical" to refer to aerosol properties. Please, fix the order of these three terms in your paper.

Line 6: I would rewrite as "a pollution period with aerosols originated in Eastern Europe"

Line 8: probably you should say: "to assess the importance of the direct radiative impact of PMA compared to other sources above the Western Mediterranean".

Introduction

Page 2, line 22: you mean "radiative forcing" or "radiative effect"? Be careful in using forcing or effect since they mean different things.

Page 2, line 23: I do not understand what do you mean with pre-existing particle loadings. Please rewrite.

Page 2, line 32-33: there are many works also in Central and Western Mediterranean characterizing the aerosol chemical, physical, and optical properties.

Page 3, line 8: I would rewrite as "the first part of this paper"

Section 2.1 Please, provide more details concerning corrections, data analysis and uncertainties for all the different used instruments. For instance, provide uncertainties

on chemical data, AERONET retrievals, nephelometer measurements. Did you correct the nephelometer for truncation? What about the correction you applied to size data? Please give more details.

Section 2.2 I would suggest rewriting line 6 as "the signals for chloride are generally lower and those for nitrate stronger for aged sea salts", otherwise it is misleading and it seems you performed a priori selection of fresh/aged PMA regardless of chemical data.

Figure 1: please add a legend indicating the species associated to the different peaks.

Section 2.4 I wonder if the aerosol mass imbalance that you find in your data is systematic or it is associated only to specific periods/days. What is the impact of this imbalance in your results? I think this is a key aspect to validate your results on the aerosol chemical composition and associated aerosol type discrimination.

Section 3. I would encourage the authors to try to reorganize a little the presentation of results/discussion in order to shorten it a little. As it is in the present form I have the impression that there are some repetitions. For instance, Section 3.2.4 and 3.3.2 could be merged and the discussion on the radiative effect and comparison between the effect of PMA/dust/pollution particles discussed in the same paragraph. Similar for the physical/optical properties paragraphs.

Section 3.1/Figure 2 Does the high nssCa2+ during the PMA period would indicate dust influence? Please check Figure 2, since some captions are missing.

Section 3.1.1 Please provide some more explanation concerning Figure 5 since it is not easy to understand.

Section 3.3.1/ Figures 10-11 See general comment.

Section 3.3.2 By Looking at the nephelometer data in Fig. 12 it seems to me that the spectral variability of the nephelometer is relatively high for a dust episode, so probably here you have the mixing of dust with smaller particles. See also general comment

regarding the representativeness of surface data.

Figure 13. I guess here you should refer to radiative effect and not to radiative forcing

---

## Author Comment (AC1) · 12 Apr 2017

**General comments :**

We thank the referees for their constructive reviews. Our replies to the two referees are given below. The main changes to the manuscript include :

- · We added a figure (Figure 2) representing the reconstructed PM10 mass.
- · We added a figure (Figure 4) representing a correlation plot between chemical components, PM1 and PM10 mass concentration and wind speed and direction.
- · Figure 3 (Figure 2 in the previous version) was modified, and two time series were added : PM1 and PM10 mass concentrations.
- · A correction for truncation was added on nephelometer scattering coefficients. Figure 13 was then modified, as well as Table 4.
- · The abstract has been shortened, with more emphasis on PMA, and the results/discussion part has been reorganized. Parts 3.2.3 and 3.3.1 have been merged, as well as Parts 3.2.4 and 3.3.2.

**Reply to referee 1**

M. Claeys et al.

We thank referee 1 for the evaluation of our manuscript. Our point-by-point responses to the comments are given below.

**Despite the fact that the instrumental set up has been described in details, providing also information for calibration and quality control where necessary, it is not clear if the MAAP was sampling through a PM1 (P.5, l. 7) or PM2.5 (P.7, l. 9) inlet. This information could be addressed along with the first reference to MAAP (P.4, l.1-3).**

The MAAP was sampling through a PM2.5 inlet, this information has been added with the first reference to MAAP (P3, l.31)

**The addition of the TEOM PM10 and TEOM PM1 plot as part of Figure 2 or as supplementary material would be substantial.**

The mass concentrations of TEOM PM1 and PM10 have been added to Figure 3 i) and j) (previously Figure 2).

**The reconstructed PM10 could also be included, since all these parameters are examined thoroughly in the manuscript (section 3.1).**

Rather than adding the reconstructed PM10 mass as a time series we present it as a scatterplot as a function of the TEOM PM10 mass concentration (Figure 2).

**Also a description of AOD at 500 nm is described (P. 15, l. 1-5) related to Figure 12a presenting the AOD temporal variability during the campaign at 440 nm and 870 nm. These two wavelengths are useful for the demonstration of the spectral dependence; nevertheless the authors could consider including the AOD time series at 500 nm also and indicate the different**

**color code in the caption of the figure.**

The caption with color code has been added for the AOD and nephelometer scattering coefficient. The AOD time series at 500 nm has also been added.

**In accordance with the previous comment, the means and standard deviation for each parameter conserning the total period could be added to the summary on Table 4.**

These values have been added on Table 4 for the ADRIMED period (total period of the field campaign).

**A diurnal variability of AE for the second part (July) of the campaign was revealed under the impact of biomass burning (P. 17-18). Nevertheless, an explanation or references of similar variability are not provided. Were the factors controlling the observed diurnal pattern investigated?**

The Angstrom exponent is highest during the day and lowest during the night, and is probably related to the diurnal cycle of the boundary layer height, as the site is situated at ~ 600 m.asl altitude.
Similar behaviors of diurnal variations of aerosol properties have previously been observed in high altitude sites (Venzac et al., 2009; Freney et al., 2011). Higher concentration of accumulation particles was recorded during daytime.
However, the number concentration of aerosols and the mass concentration of BC, organics or pollution tracers (Fig. 3 of the manuscript) do not show the same behavior. Therefore, we do not dispose of independent confirmation, and can not conclude on this diurnal variability

**Which is the contribution of nss-ions on the total ionic level overall and for each period independently? Increased nss-ions during dust and biomass burning comparatively to marine influenced period could be additionally used as indication for the presence of other sources at the site apart from marine.**

We thank the referee for this comment. Indeed, the study of the nss mass concentration compared to the total PM10 mass concentration reveals that there is an increase of nss-ions during Dust and BBP period compared to PMA period.
The contribution of nss ions to the total ionic level is 82 ± 14 % during the ADRIMED period, 92 ± 3 % during Dust period, 84 ± 5 % during BBP period, which are much higher than 53 ± 11% during PMA period.

This information has been added in the paper (p.9, l.27) :
« Furthermore, while the contribution of PMA to $PM_{10}$ mass concentration is high during PMA period, the mass contribution of nss-ions to the total ionic content is relatively low during the PMA period (53 ± 11 %). In comparison, the mass contribution of nss-ions to the total ionic content is 84 ± 5 % for the ADRIMED field campaign, and is 82 ± 14 % and 92 ± 3 % for the Dust and BBP periods respectively. Furthermore, the $Ca^{2+}$ concentration measured during the PMA
period (up to 2 μg m$^{-3}$ ) indicates the presence of dust particles, probably related to strong winds lifting soil/dust in the vicinity of the Ersa station (Arndt et al., 2017). However, unlike the Dust period, they do not represent the dominant aerosol influence during the PMA period.”

**P 14, l. 5: taking into account that the PMA is analyzed in that section it would be more appropriate to comment the low or high marine aerosol concentration instead of the presence**

**or not of marine aerosol. Unless the comment refers to all periods.**

P15, l.10 :
We have replaced « whether they contain PMA or not, aged or fresh » by « whether they contain low or high PMA concentration, aged or fresh » .

**Technical corrections**

**P. 5, l. 27-31: The information about the nephelometer is duplicated. It has already been described in pages 3-4. The comment about the scattering coefficient relation to aerosol size and concentration could be transferred in that point.**

Thank you for pointing this out. We have deleted the information on page 5 and added the following text on page 3, l.32 :
« The nephelometer provides the scattering coefficient (not directly linked to the concentration of particles), associated to an indication of the size of aerosols through the spectral dependence of the scattering coefficient between two wavelengths. The nephelometer data were corrected for truncation according to Anderson and Ogren (1998) for the total aerosol population. A correction factor of 1.29, 1.29 and 1.26 was respectively applied to the scattering coefficients at the wavelengths 450, 550 and 700 nm. »

**Figure 2: I would recommend to authors to check the plots a-k. Please pay attention on the caption and axis labels as well, especially for plots i-k. Namely: Plot i demonstrates very low wind speed. Under my opinion it is not valid. In P.10, l. 3-4 the authors refer that "At the Ersa site, during the dust outbreak, around 19 June, the wind speed reached 15 m s−1". Plot j is probably wind speed instead of wind dir (according also to figures 7 and 8, wind speed is up to 20 m s-1). Please indicate what is monitored in plot k. It seems to be wind dir. Furthermore, according to P.8, l. 23-24, BC highest concentration encountered on July 5 was equal to 0.75 μg m-3. Based on Figure 2g the maximum BC concentration was at the range of 2.5 μg m-3 (same date) or BC is actually depicted in Figure 2h. Plots e and g seem to be the same.**

We thank the reviewer for pointing out these errors in the manuscript. The axes were corrected, and PM1 and PM10 mass concentrations were added to Figure 3 (previously Figure 2).

**Typing errors:**
**P. 13, l.6: in function instead of "in fonction"**

This has been modified.

**P. 18, l.31: SW DRF at TOA is depicted in Figure 13 a, not b.**

This has been modified.

**References**

Anderson, T. L., & Ogren, J. A. (1998). Determining aerosol radiative properties using the TSI 3563 integrating nephelometer. *Aerosol Science and Technology,29*(1), 57-69.

Arndt, J., Sciare, J., Mallet, M., Roberts, G. C., Marchand, N., Sartelet, K., Sellegri, K., Dulac, F., Healy, R. M., and Wenger, J. C.: Sources and mixing state of summertime background aerosol in the northwestern Mediterranean basin, Atmos. Chem. Phys. Discuss., doi:10.5194/acp-2016-1044, in review, 2017.

Carslaw, D.C. and K. Ropkins, (2012). openair — an R package for air quality data analysis. Environmental Modelling & Software. Volume 27-28, 52-61.

Cros, B., Durand, P., Cachier, H., Drobinski, P., Frejafon, E., Kottmeier, C., Perros, P., Peuch, V.-H., Ponche, J.-L., Robin, D., et al.: The 30 ESCOMPTE program: an overview, Atmospheric Research, 69, 241–279, 2004.

Di Iorio, T., di Sarra, A., Sferlazzo, D., Cacciani, M., Meloni, D., Monteleone, F., Fua, D., and Fiocco, G.: Seasonal evolution of the tropospheric aerosol vertical profile in the central Mediterranean and role of desert dust, Journal of Geophysical Research: Atmospheres, 114, 2009.

Di Sarra, A., Di Biagio, C., Meloni, D., Monteleone, F., Pace, G., Pugnaghi, S., and Sferlazzo, D.: Shortwave and longwave radiative effects ofnthe intense Saharan dust event of 25–26 March 2010 at Lampedusa (Mediterranean Sea), Journal of Geophysical Research: Atmospheres,116, 2011.

Dubovik, O., Smirnov, A., Holben, B. N., King, M. D., Kaufman, Y. J., Eck, T. F., and Slutsker, I.: Accuracy assessment of aerosol optical properties retrieval from AERONET Sun and sky radiance measurements, J. Geophys. Res., 105, 9791–9806, doi:10.1029/2000JD900040, 2000.

Dubovik, O., Holben, B., Eck, T. F., Smirnov, A., Kaufman, Y. J.,King, M. D., Tanré, D., and Slutsker, I.: Variability of absorption and optical properties of key aerosol types observed in world-wide locations, J. Atmos. Sci., 59, 590–608, doi:10.1175/1520- 0469(2002)059<0590:VOAAOP> 2.0.CO;2, 2002.

Gantt, B., & Meskhidze, N. (2013). The physical and chemical characteristics of marine primary organic aerosol: a review. *Atmospheric Chemistry and Physics*, *13*(8), 3979-3996.

Garcıa, O. E., Dıaz, J. P., Expósito, F. J., Dıaz, A. M., Dubovik, O., Derimian, Y., ... & Roger, J. C. (2012). Shortwave radiative forcing and efficiency of key aerosol types using AERONET data. *Atmos. Chem. Phys*, *12*, 5129-5145.

Guerrero-Rascado, J. L., Olmo Reyes, F. J., Avilés-Rodríguez, I., Navas-Guzmán, E., Pérez-Ramírez, D., Lyamani, H., and Alados-Arboledas, L.: Extreme Saharan dust event over the southern Iberian Peninsula in september 2007: active and passive remote sensing from surface and satellite, Atmospheric Chemistry and Physics, 2009

Mallet, M., Dulac, F., Formenti, P., Nabat, P., Sciare, J., Roberts, G., ... & Denjean, C. (2016). Overview of the Chemistry-Aerosol Mediterranean Experiment/Aerosol Direct Radiative Forcing on the Mediterranean Climate (ChArMEx/ADRIMED) summer 2013 campaign. *Atmospheric Chemistry and Physics, 16*(2), 455-504.

Meloni, D., Di Sarra, A., Di Iorio, T., and Fiocco, G.: Direct radiative forcing of Saharan dust in the Mediterranean from measurements at Lampedusa Island and MISR space-borne observations, Journal of Geophysical Research: Atmospheres, 109, 2004.

Pey, J., Querol, X., and Alastuey, A.: Variations of levels and composition of PM10 and PM2.5 at an insular site in the Western Mediterranean, Atmos. Res., 94, 285–299, doi:10.1016/j.atmosres.2009.06.006, http://dx.doi.org/10.1016/j.atmosres.2009.06.006, 2009.

Sellegri, K., Gourdeau, J., Putaud, J.-P., and Despiau, S.: Chemical composition of marine aerosol in a Mediterranean coastal zone during the FETCH experiment, J. Geophys. Res., 106, 12 023, doi:10.1029/2000JD900629, 2001.

Tang, I. N., 1997 : Thermodynamic and optical properties of mixed-salt aerosols of atmospheric importance. Journal of Geophysical Research : Atmospheres, 102 (D2), 1883–1893.

---

## Author Comment (AC2) · 12 Apr 2017

**General comments :**

We thank the referees for their constructive reviews. Our replies to the two referees are given below. The main changes to the manuscript include :

- · We added a figure (Figure 2) representing the reconstructed PM10 mass.
- · We added a figure (Figure 4) representing a correlation plot between chemical components, PM1 and PM10 mass concentration and wind speed and direction.
- · Figure 3 (Figure 2 in the previous version) was modified, and two time series were added : PM1 and PM10 mass concentrations.
- · A correction for truncation was added on nephelometer scattering coefficients. Figure 13 was then modified, as well as Table 4.
- · The abstract has been shortened, with more emphasis on PMA, and the results/discussion part has been reorganized. Parts 3.2.3 and 3.3.1 have been merged, as well as Parts 3.2.4 and 3.3.2.

**Reply to referee 2**

M. Claeys et al.

We thank referee 2 for the evaluation of our manuscript. Our point-by-point responses to the comments are given below.

**1/ you do not reach mass closure between TEOM data and PILS/MAAP/ACSM (Section 2.4) and I wonder what is the impact of this on your results. Is the aerosol chemical composition that you find representative of the whole aerosol population, or not? This is a key aspect to validate your results on the aerosol composition and associated aerosol type discrimination.**

We believe there is no impact on the results. The three main aerosol types presented in this manuscript have been determined assimilating multiple independent sources of information, including key chemical tracers (measured by the PILS and ACSM), optical properties (MAAP and nephelometer), along with FLEXPART back trajectory analysis. In addition, it is highly unlikely for any source to have emitted completely undetectable compounds for the suite of instruments used here.

For example (as described in Section 3.1), the inorganic PMA (PM10) concentration represents at least 40 % of the $PM_{10}$ mass concentration (TEOM) during the PMA period, which is independent of the reconstructed mass. We can also clearly identify the PMA period by its mass concentration (sea salt), which is higher (reaching 6 µg m$^{-3}$) during these days compared to the rest of the field campaign when it does not exceed 1 µg m$^{-3}$. Similarly, the Dust period is detected by the presence of Calcium at the Ersa station, which is a common tracer of dust aerosols, and by airplane measurements (Mallet et al., 2016). The BBP period is also identified by the presence of potassium, BC, and higher $PM_1$ mass concentration.

In addition, FLEXPART back trajectories confirm the origin of the airmasses. The presence of dust particles is supported by the African / Saharan origin of air masses, as well as for the BBP period, with an origin from East Europe (particularly Russia).  PMA concentrations were also highest during periods of strongest winds – related to direct emissions of PMA.

The ratio of the reconstructed mass over the TEOM PM10 mass concentration is lower during the PMA period (0.65 ± 0.20) compared to the BBP period (0.74 ± 0.23) and we do not dispose of the TEOM PM10 data during the DUST period.
The three instruments used for this analysis have some constraints. Indeed the PILS instrument measures the water-soluble material, the MAAP instrument the absorbing aerosol, and the ACSM the non-refractory compounds. Therefore, there are some losses compared to the total aerosols mass concentration, but the losses do not constitute sources by themeselves.

Indeed, there are some losses in the PILS-IC, due to the sampling lines and to the solubilization process. For example, Ca2+ is not highly soluble so it could contribute to the underestimation by the reconstructed mass.
It may also be due to the fact that the organic aerosols were sampled through a $PM_1$ inlet. Even though they are mainly in the submicronic part, we may miss a mass contribution to the $PM_{10}$. Gantt and Meskhidze (2013) summarized reasults of measurements of the organic mass fraction of sea spray aerosol in function of their size. For the supermicronic (between 1 -2.5 µm) part of the spectrum, the mass contribution of organics can still represents more than 10 % of sea spray. These organics would not be detected by any of the techniques used during ADRIMED campaign and could explain the higher missing mass during PMA.

We do not find any significant correlation between missing mass and TEOM PM10 mass, even though the missing mass ratio is globally higher when the PM10 total mass is higher (see next figure).

[Figure]

*Figure 1: TEOM PM10 mass concentration (ug m-3) in function of the missing mass ratio (%)*
In section 2.4, the following sentence has been replaced by the following paragraph (p.7, l.11):

« We note that the reconstructed mass underestimates the TEOM PM10 concentration by a factor

ranging from 0.5 to 1 with a poor correlation coefficient (r2=0.31), as illustrated in Figure 2. »

« The ratio of the reconstructed mass over the TEOM PM10 mass concentration average 0.79 during the ADRIMED campaign. It is lower during the PMA period (0.65 ± 0.20) compared to the BBP period (0.74 ±0.23); we did not measure TEOM PM10 during the DUST period. We do not find any significant correlation between missing mass and TEOM PM10 mass, even though the missing mass is globally higher when the PM10 total mass is higher.

This lack of aerosol mass could also be due to the mass of (insoluble) dust not determined chemically or possibly a supermicron mode of organics that was not determined here. Indeed, the organic mass fraction can represent more than 10 % of the sea spray mass for aerosols comprised between 1 and 3 µm during periods of high biological activity (Gantt and Meskhidze, 2013), this ratio decreasing with increasing sizes.

Even though full mass closure has not been reached, there is no impact on the results because the losses do not represent sources by themselves. The three main aerosol types presented in this manuscript have been determined using key chemical tracers (measured by the PILS and ACSM), optical properties (MAAP and nephelometer), FLEXPART back trajectory analysis for confirmation. The combination of these different analyses conducted in this study is found to be coherent and representative of the whole aerosol population. »

**2/ By looking at Figure 10 I would expect larger differences in the size distribution for the three cases, especially in the coarse part. Instead, size distributions seem to agree within uncertainties for PMA, dust, and pollution/biomass burning aerosols. How can you explain this? For dust, this is due to the fact that, as you say in the paper, particles are mostly located above 3 km, while your measurements are at the ground. On the contrary, when you look at column averaged data (Figure 11), you have very large differences in the size distribution for the three periods. This is reasonable since AERONET data represent atmospheric condition over the whole column. By looking at these two plots, however I wonder how representative surface data are and how well can they be used to correctly discriminate between the three periods. This is a crucial point to validate the results/observations at the surface.**

As stated above, the three main aerosol types presented in this manuscript have been determined assimilating multiple independent sources of information. We investigated the size distribution during the three periods to see if we could discriminate the major aerosol influence. This study shows that the size distribution at the surface is not enough to determine the different aerosol regimes, and that chemical composition is a necessary information to discriminate these three periods.

The contribution of anthropogenic aerosols can explain this result. Indeed, the number concentration of submicronic particles was always relatively high, similar to an urban background site. Furthermore, the relative contribution of PMA and long range transport (dust and biomass burning aerosols) are relatively small.

We attribute the relatively small concentration of coarse PMA particles in Ersa to dry deposition as the station is situated at almost 600 m asl (often at the top of the marine boundary layer)

Concerning the dust event, its amplitude was relatively low above Corsica (AOD reaching 0.3 compared to values above 1 for large dust outbreaks (Guerrero-Rascado et al., 2009; Di Sarra et al., 2011) in the Central and Western Mediterranean). In addition, the main dust layers are transported

in the free troposphere well above the measurements at Ersa (Denjean et al., 2016). We also note on Figure 11 that the highest volume size concentration in the highest diameters (4 to 10 µm) corresponds to the Dust period.

As mentioned in the previous comment, FLEXPART back-trajectories confirm the origin of the air masses and corroborate aerosol chemistry measurements for each period, dust, primary marine aerosols and biomass burning mixed with anthropogenic aerosols.

To illustrate, Figure 2 represents a correlation plot of the different chemical component mass concentration, PM1 and PM10 mass concentration as well as wind speed and direction, to help visualise relationships between variables. In this figure, the order of the variables appear due to their similarity with one another, through hierarchical cluster analysis (Carslaw, D.C. and K. Ropkins, 2012). The color and the number represent the correlation between two variables, when close to 100, the correlation is high. The shape of the ellipse is a visual representation of a scatterplot. We can observe on this figure two groups of variables. The first one composed of Cl, Na, Ca, K and PM10 mass concentration, related to marine or terrestrial influence, while the second one, composed of NH4, SO41, BC, Organics and PM1 mass concentration, is related to pollution influence.

[Figure]

*Figure 2: Correlation plot of chemical constituents mass concentrations, PM1 and PM10 mass concentration and wind speed and direction, during the whole campaign.*

The next paragraph was added on the manuscript (p.8, l.9 ) :

"A correlation plot (Fig.4) illustrates the relationship between the principal chemical constituents, $PM_1$ and $PM_{10}$ mass concentrations, as well as wind speed and direction. In the figure, the order of the variables appear due to their similarity with one another, through hierarchical analysis (Carslaw, D.C. and K. Ropkins, 2012) . The color and the number represent the correlation between two variables, when close to 100, the correlation is high. The shape of the ellipse is a visual representation of a scatterplot. We can observe on this figure two groups of variables. The first one composed of Cl, Na, Ca, K and PM10 mass concentration, related to marine or terrestrial influence, while the second one, composed of NH4, SO41, BC, Organics and PM1 mass concentration, is related to pollution influence."

**Specific comments**
**Abstract I would suggest the authors to partly rewrite the abstract to put more in evidence the role of marine aerosols, since in the present form it seems to me not fully in line with the title/text of the paper. It seems to me that the accent is put mostly on the estimate of the direct radiative effect of sea salt compared to dust and poll tion/biomass burning, while this aspect represent only a part of the paper. I would also suggest adding a sentence at the end of the abstract to highlight your conclusions. Also, but this is a minor thing, throughout the abstract and the paper you use randomly "optical, physical, chemical", "physical, optical, chemical", or "chemical, physical, optical" to refer to aerosol properties. Please, fix the order of these three terms in your paper.**

The abstract has been modified.
The three terms have been fixed to « Optical, physical and chemical », like the title.

**Line 6: I would rewrite as "a pollution period with aerosols originated in Eastern Europe"**

« a pollution period from Eastern Europe » has been replaced by "a pollution period with aerosols originating from Eastern Europe"

**Line 8: probably you should say: "to assess the importance of the direct radiative impact of PMA compared to other sources above the Western Mediterranean".**

« to assess the direct radiative impact of PMA above the Western Mediterranean Basin » has been replaced by «to assess the importance of the direct radiative impact of PMA compared to other sources above the Western Mediterranean »

**Introduction**
**Page 2, line 22: you mean "radiative forcing" or "radiative effect"? Be careful in using forcing or effect since they mean different things.**

Indeed, we do mean « radiative forcing » (Bellouin et al., 2008).

**Page 2, line 23: I do not understand what do you mean with pre-existing particle loadings. Please rewrite.**

« pre-existing particle loadings » has been replaced by « long-range transport of marine aerosols »

**Page 2, line 32-33: there are many works also in Central and Western Mediterranean characterizing the aerosol chemical, physical, and optical properties.**

Page 2, lines 25-26:
Some references were added for Central (Meloni et al., 2004 , Di Iorio et al., 2009) and Western Mediterranean (Sellegri et al., 2001, Cros et al., 2004, Pey et al., 2009, Guerrero-Rascado et al., 2009)

**Page 3, line 8: I would rewrite as "the first part of this paper"**

« The first part of this study » has been replaced by « The first section of this manuscript »

**Section 2.1 Please, provide more details concerning corrections, data analysis and uncertainties for all the different used instruments. For instance, provide uncertainties on chemical data, AERONET retrievals, nephelometer measurements. Did you correct the nephelometer for truncation? What about the correction you applied to size data? Please give more details.**

The accuracy of AERONET retrievals are discussed by Dubovik et al. (2000, 2002) and this sentence was added to the text (p.5, l.29) :
"The accuracy of AERONET retrievals are discussed by Dubovik and King (2000); Dubovik et al. (2002a)".

The nephelometer data were not previously corrected for truncation, so Figure 13 and Table 4 now takes into account this correction. The nephelometer data were corrected for truncation according to Anderson and Ogren (1998) method. We used only the total scattering; therefore, no discrimination was made between sub-micron and super-micron aerosol scattering during the measurements. The truncation errors associated to the total aerosol population at three wavelengths are:

|  | 450 nm | 550 nm | 700 nm |
|---|---|---|---|
| No size cut | $1.29 \pm 0.23$ | $1.29 \pm 0.23$ | $1.26 \pm 0.21$ |

*(Based on Anderson and Ogren (1998))*

We added this sentence to the text (p.4, l.1):
« The nephelometer data are corrected for truncation according to Anderson and Ogren (1998) method for the total aerosol population. A correction factor of 1.29, 1.29 and 1.26 is applied to the scattering coefficients at the wavelengths 450, 550 and 700 nm, respectively. »

As stated in the manuscript, for the measurement of their size distributions, aerosol had been dried to RH < 40 %. We assume a shape factor equal to 1.

The size distributions from the SMPS have been corrected using the standard techniques within the TSI software (diffusion losses, charge distribution, multiple charges). We directly present OPS size distributions with no further corrections.

We also estimate the impact of the optical properties (refractive index and absorption) on the OPS signal using a Mie code, and taking into account the optical geometry and the laser wavelength of the OPS instrument. We use references values for the refractive indexes of dust aerosols, primary marine aerosols and biomass burning / pollution aerosols. The results are presented in the next figure. The calibration of the OPS instrument was done with the refractive index of PSL, equal to $1.59 - 0.000\ i$.

The following optical properties were chosen for each type of aerosol:

|  | PSL (calibration) | Dust | PMA | BB |
|---|---|---|---|---|
| Refractive index | $1.59 - 0.000\ i$ | $1.52 - 0.002\ i$ | $1.54 + 0.000\ i$ | $1.53 - 0.007\ i$ |

As can be seen on the next figure, the difference in the size distribution for dust and primary marine aerosols compared to the OPS refractive index for calibration is not significant (< 10% difference is section efficiency for NaCl and Dust aerosol < 2.5 um diameter).

For the BB aerosols, as they are mainly submicron, the impact of a correction of the refractive index is also negligible.

[Figure]

*Figure 3: Theoretical impact of different refractive indexes on the response signal of the OPS (scattering cross section) in function of the particles diameter. Calculated by using a Mie code.*

This sentence was added to the text (p.3, l.23) :
"Optical corrections to the OPS size distributions are negligible when accounting for the refractive indices associated to the different particle types. We considered the particles as spherical (shape factor equal to 1)."

**Section 2.2 I would suggest rewriting line 6 as "the signals for chloride are generally lower and those for nitrate stronger for aged sea salts", otherwise it is misleading and it seems you performed a priori selection of fresh/aged PMA regardless of chemical data.**

We agree: « The signals for chloride were lower and those for nitrate stronger for aged sea salt » has been replaced by « the signals for chloride are generally lower, and those for nitrate stronger for aged sea salts »

**Figure 1: please add a legend indicating the species associated to the different peaks.**

The legend has been added to Figure 1.

**Section 2.4 I wonder if the aerosol mass imbalance that you find in your data is systematic or it is associated only to specific periods/days. What is the impact of this imbalance in your results? I think this is a key aspect to validate your results on the aerosol chemical composition and associated aerosol type discrimination.**

The aerosol mass imbalance fluctuates as a function of time, according to the following figure, which represents the ratio of the reconstructed $PM_{10}$ mass concentration over the $PM_{10}$ mass concentration.

The TEOM $PM_{10}$ measurements are not available over the entire campaign, particularly during the Dust period. During the ADRIMED campaign, the ratio of the rebuilt PM10 mass concentration to the TEOM PM10 mass concentration is 0.79. During PMA period, this ratio is 0.65 ± 0.20, while it is 0.74 ± 0.23 during the BBP period.

As discussed previously in response 1, this mass imbalance does not influence the aerosol type classification.

[Figure]

*Figure 4: Ratio of the rebuilt PM10 mass over TEOM PM10 mass concentration during the ADRIMED campaign*

**Section 3. I would encourage the authors to try to reorganize a little the presentation of results/discussion in order to shorten it a little. As it is in the present form I have the impression that there are some repetitions. For instance, Section 3.2.4 and 3.3.2 could be merged and the discussion on the radiative effect and comparison between the effect of PMA/dust/pollution particles discussed in the same paragraph. Similar for the physical/optical properties paragraphs.**

We thank the reviewer for this comment. The results/discussion part has been reorganized. Parts 3.2.3 and 3.3.1 has been merged, as well as parts 3.2.4 and 3.3.2.

**Section 3.1/Figure 2 Does the high nssCa2+ during the PMA period would indicate dust**

**influence? Please check Figure 2, since some captions are missing.**

Indeed, the nss-Ca2+ mass concentration reaches 2 µg m$^{-3}$ during the PMA period so there is an influence of dust particles, but from local origin (Arndt et al, 2017) – related to the strong winds lifting soil / dust near the Ersa Station.

Comments added on the section 3.1 (p.9 l.30):
"Furthermore, the Ca$^{2+}$ concentration measured during the PMA period (up to 2 µg m$^{-3}$) indicates the presence of dust particles, probably related to strong winds lifting soil/dust in the vicinity of the Ersa Station (Arndt et al, 2017). However, unlike the Dust period, they do not represent the dominant aerosol influence during the PMA period."

Figure 3 (Figure 2 in the previous version) has been modified, the missing caption added and two time series have been added : PM$_1$ and PM$_{10}$ mass concentrations.

**Section 3.1.1 Please provide some more explanation concerning Figure 5 since it is not easy to understand.**

Comments added on the section 3.1.1 (p.10 l.11) :
« For each day during the campaign (bottom axes), the upper figure indicates the different zones through which the air masses passed before reaching Ersa. The bottom figure indicates the transport time from these zones to the Ersa sample site. »

**Section 3.3.1/ Figures 10-11 See general comment.**

The results of figures 12-13 (previously 10-11) are discussed in the answer of the second general comment.

**Section 3.3.2 By Looking at the nephelometer data in Fig. 12 it seems to me that the spectral variability of the nephelometer is relatively high for a dust episode, so probably here you have the mixing of dust with smaller particles. See also general comment regarding the representativeness of surface data.**

The number size distribution at the surface shows indeed the presence of accumulation particles related to anthropogenic sources (d=130 nm) during the Dust period. As stated in the manuscript, Ersa station is situated in a continental rural background site, with relatively high background number concentration of a few thousands particles.

A comment was added on the paper (p.16 l.18) :
« even though a fine mode is also detected during dust period. »
And (p.18 l.17) :
« PMA period is characterized by a relatively weak wavelength dependency (Fig. 13 c).  While the mixing of dust with fine particles, previously shown by the AERONET volume size distribution, is shown here by a relatively high wavelength dependency (mean of 20 ± 9) »

**Figure 13. I guess here you should refer to radiative effect and not to radiative forcing**

We thank the reviewer for catching this error.  Indeed, the terms « radiative forcing » have been replaced by « radiative effect ».

**References**

Anderson, T. L., & Ogren, J. A. (1998). Determining aerosol radiative properties using the TSI 3563 integrating nephelometer. *Aerosol Science and Technology,29*(1), 57-69.

Arndt, J., Sciare, J., Mallet, M., Roberts, G. C., Marchand, N., Sartelet, K., Sellegri, K., Dulac, F., Healy, R. M., and Wenger, J. C.: Sources and mixing state of summertime background aerosol in the northwestern Mediterranean basin, Atmos. Chem. Phys. Discuss., doi:10.5194/acp-2016-1044, in review, 2017.

Carslaw, D.C. and K. Ropkins, (2012). openair — an R package for air quality data analysis. Environmental Modelling & Software. Volume 27-28, 52-61.

Cros, B., Durand, P., Cachier, H., Drobinski, P., Frejafon, E., Kottmeier, C., Perros, P., Peuch, V.-H., Ponche, J.-L., Robin, D., et al.: The 30 ESCOMPTE program: an overview, Atmospheric Research, 69, 241–279, 2004.

Di Iorio, T., di Sarra, A., Sferlazzo, D., Cacciani, M., Meloni, D., Monteleone, F., Fua, D., and Fiocco, G.: Seasonal evolution of the tropospheric aerosol vertical profile in the central Mediterranean and role of desert dust, Journal of Geophysical Research: Atmospheres, 114, 2009.

Di Sarra, A., Di Biagio, C., Meloni, D., Monteleone, F., Pace, G., Pugnaghi, S., and Sferlazzo, D.: Shortwave and longwave radiative effects ofnthe intense Saharan dust event of 25–26 March 2010 at Lampedusa (Mediterranean Sea), Journal of Geophysical Research: Atmospheres,116, 2011.

Dubovik, O., Smirnov, A., Holben, B. N., King, M. D., Kaufman, Y. J., Eck, T. F., and Slutsker, I.: Accuracy assessment of aerosol optical properties retrieval from AERONET Sun and sky radiance measurements, J. Geophys. Res., 105, 9791–9806, doi:10.1029/2000JD900040, 2000.

Dubovik, O., Holben, B., Eck, T. F., Smirnov, A., Kaufman, Y. J.,King, M. D., Tanré, D., and Slutsker, I.: Variability of absorption and optical properties of key aerosol types observed in world-wide locations, J. Atmos. Sci., 59, 590–608, doi:10.1175/1520- 0469(2002)059<0590:VOAAOP> 2.0.CO;2, 2002.

Gantt, B., & Meskhidze, N. (2013). The physical and chemical characteristics of marine primary organic aerosol: a review. *Atmospheric Chemistry and Physics, 13*(8), 3979-3996.

Garcıa, O. E., Dıaz, J. P., Expósito, F. J., Dıaz, A. M., Dubovik, O., Derimian, Y., ... & Roger, J. C. (2012). Shortwave radiative forcing and efficiency of key aerosol types using AERONET data. *Atmos. Chem. Phys, 12*, 5129-5145.

Guerrero-Rascado, J. L., Olmo Reyes, F. J., Avilés-Rodríguez, I., Navas-Guzmán, E., Pérez-Ramírez, D., Lyamani, H., and Alados-Arboledas, L.: Extreme Saharan dust event over the southern Iberian Peninsula in september 2007: active and passive remote sensing from surface and satellite, Atmospheric Chemistry and Physics, 2009

Mallet, M., Dulac, F., Formenti, P., Nabat, P., Sciare, J., Roberts, G., ... & Denjean, C. (2016). Overview of the Chemistry-Aerosol Mediterranean Experiment/Aerosol Direct Radiative Forcing on the Mediterranean Climate (ChArMEx/ADRIMED) summer 2013 campaign. *Atmospheric*

*Chemistry and Physics*, *16*(2), 455-504.

Meloni, D., Di Sarra, A., Di Iorio, T., and Fiocco, G.: Direct radiative forcing of Saharan dust in the Mediterranean from measurements at Lampedusa Island and MISR space-borne observations, Journal of Geophysical Research: Atmospheres, 109, 2004.

Pey, J., Querol, X., and Alastuey, A.: Variations of levels and composition of PM10 and PM2.5 at an insular site in the Western Mediterranean, Atmos. Res., 94, 285–299, doi:10.1016/j.atmosres.2009.06.006, http://dx.doi.org/10.1016/j.atmosres.2009.06.006, 2009.

Sellegri, K., Gourdeau, J., Putaud, J.-P., and Despiau, S.: Chemical composition of marine aerosol in a Mediterranean coastal zone during the FETCH experiment, J. Geophys. Res., 106, 12 023, doi:10.1029/2000JD900629, 2001.

Tang, I. N., 1997 : Thermodynamic and optical properties of mixed-salt aerosols of atmospheric importance. Journal of Geophysical Research : Atmospheres, 102 (D2), 1883–1893.